

# A Saddle-Node Bifurcation is Causing the AMOC Collapse in the Community Earth System Model

René M. van Westen[1], Elian Vanderborght[1], and Henk A. Dijkstra[1]

[1]Institute for Marine and Atmospheric research Utrecht, Department of Physics, Utrecht University, Utrecht, the Netherlands

**Correspondence:** René M. van Westen (r.m.vanwesten@uu.nl)

**Abstract.** Recently, a collapse of the Atlantic Meridional Overturning Circulation (AMOC) was found in the Community Earth System Model (CESM) under constant pre-industrial greenhouse gas forcing conditions. To determine the stability changes of the AMOC with changing (freshwater) parameters in models, it is important to determine the origin of the collapse behavior. In this paper, we argue that the classical picture of a saddle-node bifurcation holds for the AMOC collapse in the CESM. We provide specific supporting arguments by showing results of additional pre-industrial CESM simulations and by comparison with a conceptual model. Theoretical arguments are also provided showing that the essential dynamics of the CESM can be reduced to a low-dimensional model in which a saddle-node bifurcation causes the AMOC collapse. The underlying physical reason is that the AMOC behaviour in CESM is controlled by a small set of dominant feedback processes. This has important consequences for the value of conceptual AMOC models, for assessing the effect of model biases on the AMOC stability, and for the interpretation of the AMOC behaviour under climate change scenario's.

## 1  Introduction

A hot issue in current climate research is the Atlantic Meridional Overturning Circulation (AMOC) response under future climate change. Climate models participating in the Coupled Model Inter-comparison Project Phase 6 (CMIP6) indicate a substantial AMOC weakening during the 21$^{st}$ century (Weijer et al., 2020). Beyond 2100 there is much more uncertainty as the AMOC may (partially) recover (Bonan et al., 2022) or fully collapse (Liu et al., 2017). Transient temperature responses are effective in causing the 21$^{st}$ century AMOC weakening but salinity responses are crucial in further destabilizing the AMOC (Gérard and Crucifix, 2024; van Westen et al., 2024c). The dominant destabilizing AMOC tipping mechanism is the salt-advection feedback, where an AMOC weakening leads to a smaller northward salinity transport amplifying the initial AMOC weakening (Marotzke, 2000). The existence of the salt-advection feedback is the reason to label the AMOC as a tipping element in the climate system (Lenton et al., 2008; Armstrong McKay et al., 2022).

Stommel (1961) was the first to identify the salt-advection feedback in a simple two-box model and demonstrated that this feedback induces transitions between two stable AMOC steady states. The multi-stable AMOC regime is bounded by two saddle-node bifurcations in this model. Since then, studies using more detailed conceptual (box) models (Cessi, 1994; Cimatoribus et al., 2014) and numerically fully-implicit ocean-climate models (De Niet et al., 2007; Toom et al., 2012; Mulder et al., 2021) have shown that saddle-node bifurcations bound the multi-stable regime of the AMOC in these models. Rahmstorf





(1996) showed that the saddle-node bifurcation associated with the AMOC collapse is linked to a critical value of the freshwater transport carried by the AMOC at 34°S, represented by the quantity $F_{ovS}$. When including the stabilizing gyre responses (Sijp, 2012), a $F_{ovS}$ minimum is found close to this saddle-node bifurcation (Dijkstra, 2007).

In numerically explicit ocean-climate models it is much harder (or not feasible) to determine the steady states versus (freshwater forcing) parameters and the boundaries of the AMOC multi-stable regime. An impression of the multi-stable regime can be obtained by performing quasi-equilibrium simulations, where a freshwater flux forcing is changed very slowly back-and-forth such that the model state stays close to the (slowly changing) statistical equilibrium. Such quasi-equilibrium simulations have been performed with many ocean-only models (Rahmstorf, 1995; Lohmann et al., 2024), Earth System Models of Intermediate Complexity (EMICs) (Rahmstorf et al., 2005; Cini et al., 2024), the FAMOUS model (Hawkins et al., 2011), the Community Climate System Model (CCSM3) (Hu et al., 2012), and recently in the Community Earth System Model (CESM) (van Westen and Dijkstra, 2023; van Westen et al., 2024a).

We focus here on the latter result for the CESM and address the issue whether also this behavior is caused by the presence of a saddle-node bifurcation, similar to that in the fully-implicit ocean-climate models (Dijkstra, 2007). This is certainly a non-trivial issue as the CESM is an extremely high-dimensional dynamical system and the atmospheric fluxes create a high frequency forcing on the ocean component of the model. In addition, in the quasi-equilibrium CESM simulation (van Westen et al., 2024a) the forcing rate is rather large compared to the equilibration time scale of the AMOC (van Westen et al., 2024b) and hence the (non-autonomous) dynamical system is not a fast-slow system (Kuehn, 2011). The existence of a saddle-node bifurcation in the CESM is important for assessing the role of model biases on the stability of the AMOC and for understanding the response of the model to transient climate change forcing (Ritchie et al., 2021).

The aim of this paper is to provide a convincing case that a saddle-node bifurcation is causing the AMOC collapse in the CESM, as presented in van Westen et al. (2024a). Thereto, we have performed several additional CESM simulations which were branched from the quasi-equilibrium CESM simulation, we will compare the CESM behavior with that of a five-box model for which a saddle-node bifurcation is known to exist (van Westen et al., 2024b), and we provide additional theoretical analysis. Section 2 describes the model set-up and simulations for the CESM and five-box model and is followed in Section 3 by results on the (statistical) steady states and quasi-equilibrium results of both models. Section 4 provides detailed theoretical arguments for the existence of a saddle-node in the CESM and in Section 5, the importance of this result for the behavior of the AMOC under climate change is shown. Finally, in Section 6, the results are summarized and discussed.

## 2 Models and Methods

### 2.1 CESM simulations

The CESM (version 1.0.5) is a fully-coupled climate model and the simulations here have a 1° horizontal resolution for the ocean/sea-ice components and a 2° horizontal resolution for the atmosphere/land components. For more details on the precise CESM set-up, we refer to van Westen and Dijkstra (2023) and van Westen et al. (2024a). In those studies, the pre-industrial forcing is used and in addition a freshwater flux forcing ($F_H$) is applied between 20°N and 50°N in the Atlantic Ocean and is





compensated elsewhere (at the ocean surface) to conserve salinity. The AMOC hysteresis simulation (van Westen and Dijkstra,
2023) is obtained by slowly increasing $F_H$ from 0 Sv to 0.66 Sv and back to 0 Sv, at a rate of $3 \times 10^{-4}$ Sv yr$^{-1}$, resulting in a
4400-year long simulation.

This quasi-equilibrium simulation remains close to the statistical equilibria, but the deviations become larger near the AMOC
collapse and recovery (van Westen et al., 2024b). To determine statistical steady states, two 500-year long CESM simulations
were performed (van Westen et al., 2024b) at constant $F_H$, the steady states are indicated as $\overline{F_H}$. This was already done for
$\overline{F_H} = 0.18$ Sv (starting at model year 600 of the quasi-equilibrium simulation) and at $\overline{F_H} = 0.45$ Sv forcing (starting at model
year 1500). Below, we will show results of new CESM simulations performed under constant $\overline{F_H}$ forcing or with a slower rate
of $F_H$, and closer to the values where the AMOC collapse occurs in the quasi-equilibrium simulation (around $F_H = 0.525$ Sv,
van Westen et al. (2024a)).

We will (in Section 5) also use results from two climate change simulations with the CESM that were initialized from the
end of the statistical steady state with constant $\overline{F_H} = 0.18$ Sv and $\overline{F_H} = 0.45$ Sv. These climate change simulations were first
forced under the historical forcing (1850 – 2005) and followed by either RCP4.5 or RCP8.5 scenario forcing (2006 – 2100,
Representative Concentration Pathway). Subsequently, they were further integrated for 400 years under their 2100 radiative
forcing conditions to study the equilibrium behaviour. More details on these simulations are provided elsewhere (van Westen
et al., 2024c).

## 2.2 The 5-box ocean model

The five-box ocean model (Figure 1) was developed by Cimatoribus et al. (2014), extended by Castellana et al. (2019), and was
recently further extended (hereafter the E-CCM) by including oceanic temperatures (van Westen et al., 2024b). The AMOC
strength in the northern box ($q_N$) in the E-CCM is given by:

$$q_N = \eta_h \frac{\rho_n - \rho_{ts}}{\rho_0} D^2,\tag{1}$$

where $\eta_h$ is a hydraulic constant, $\rho_n - \rho_{ts}$ is the meridional density difference between box n and box ts, $\rho_0$ is a reference
density, and $D$ the pycnocline depth. The densities are determined from a linear equation of state. For full details and sensitivity
experiments conducted with the E-CCM, we refer to van Westen et al. (2024b), where there is also a link to the publicly-
available E-CCM code. We will show results for the version where sea-ice insulation effects are omitted and use the standard
values of the parameters given in van Westen et al. (2024b), unless otherwise mentioned.

The E-CCM is forced through the asymmetric freshwater flux forcing ($E_A$) from box s to box n. Under varying $E_A$, the
E-CCM has an 'AMOC on' state (clockwise circulation, red solid and dashed arrows) and an 'AMOC off' state (anti-clockwise
circulation, red solid and dotted arrows). There is a multi-stable AMOC regime and this regime is bounded by two saddle-node
bifurcations (van Westen et al., 2024b). To determine the sensitivity of the AMOC behavior to the hosing location (Rahmstorf,
1996; Ma et al., 2024), we make a modification to the E-CCM by distributing the freshwater flux forcing linearly over box n
and box t using a parameter $\xi \in [0, 1]$. When $\xi = 0$, the freshwater flux forcing is only applied to box n and this is the original
E-CCM configuration. The freshwater flux forcing is only over box t when $\xi = 1$.





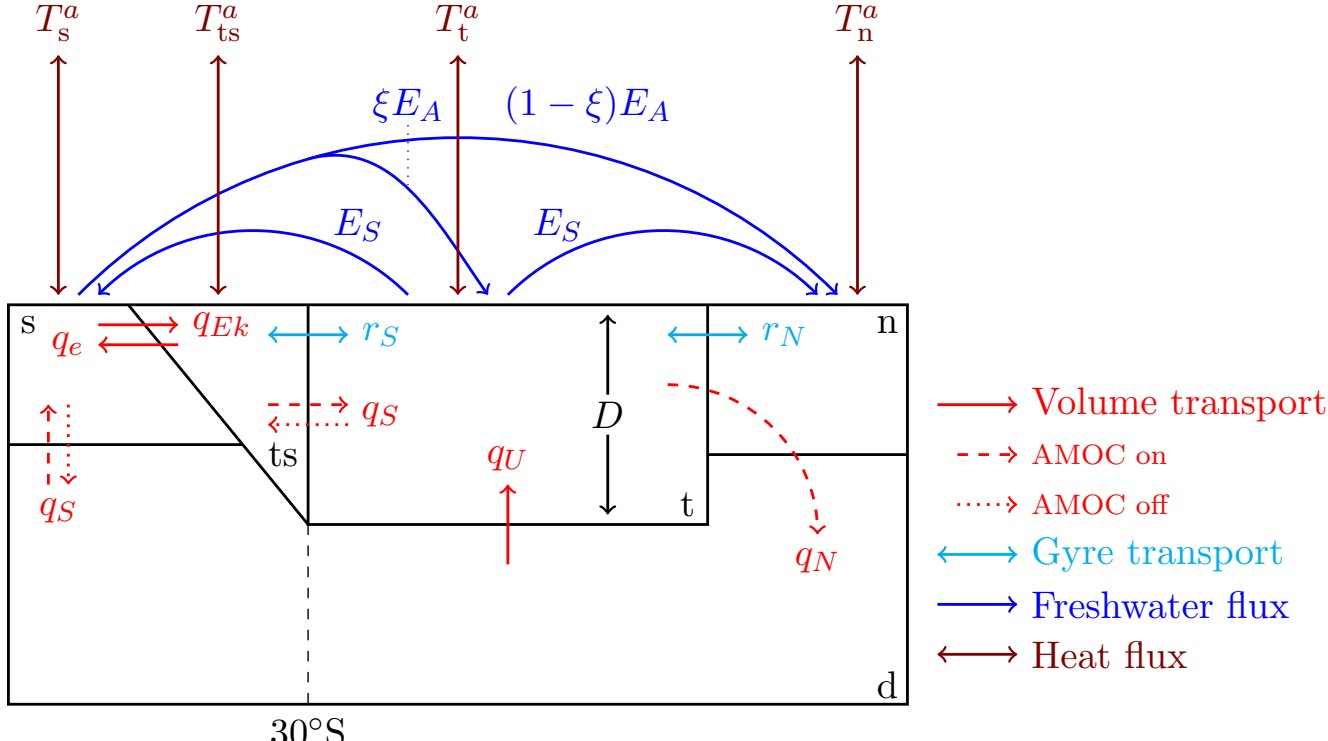

**Figure 1.** Schematic representation of the five-box AMOC model (the E-CCM), adapted from van Westen et al. (2024b). The red arrows represent volume transports, whereas the dashed and dotted arrows indicate the AMOC on and AMOC off states, respectively. The cyan and blue arrows represent the gyre transport and freshwater fluxes, respectively. The freshwater from box s is distributed linearly over box n and box t using a parameter $\xi$, where $\xi E_A$ is added to box $t$ and $(1-\xi)E_A$ to box n. The original E-CCM configuration van Westen et al. (2024b) is obtained when $\xi = 0$. The brown arrows are the heat fluxes with the overhead atmosphere for each surface box (i.e., box s, ts, t and n).

The steady states of the E-CCM against varying parameters (i.e., bifurcation diagram), such as freshwater flux forcing, are determined using the continuation software AUTO-07p (Doedel et al., 2007, 2021). This code solves steady states using a pseudo-arclength continuation combined with a Newton-Raphson method (Wubs and Dijkstra, 2023). It is also able to detect Hopf bifurcations and saddle-node bifurcations. We used a value of $10^{-6}$ for the absolute and relative accuracy of each steady-state solution, and for the accuracy for locating special points, similar to van Westen et al. (2024b).





# 3 Results

## 3.1 Statistical equilibria in the CESM

The AMOC strength (at 1,000 m and 26°N) and the freshwater transport carried by the AMOC at 34°S ($F_{\mathrm{ovS}}$) of the quasi-
equilibrium CESM simulation (van Westen et al., 2024a) are shown in Figures 2a,b. The branched simulations from the quasi-
equilibrium simulation at a constant forcing $\overline{F_H} = 0.18$ Sv (Figures 2c,i), $\overline{F_H} = 0.45$ Sv (Figures 2d,j) and $\overline{F_H} = 0.465$ Sv
(Figures 2e,k) equilibrate after about 300 years. The branched simulation at $\overline{F_H} = 0.48$ Sv (Figures 2f,l) collapses and sug-
gests that the upper bound of the multi-stable regime is around this $\overline{F_H}$ value. The branches initiated from $\overline{F_H} = 0.495$ Sv
(Figures 2g,m) and $\overline{F_H} = 0.51$ Sv (Figures 2h,n) also collapse; these simulations were terminated before the 500-year mark
because of computational costs. However, when the equilibrated $\overline{F_H} = 0.465$ Sv simulation is subjected to an instantaneous
increase in freshwater flux to $\overline{F_H} = 0.48$ Sv, we still find a statistical equilibrium in the northward overturning regime (red
curves in Figures 2f,l). We iteratively repeated the same procedure for $\overline{F_H} = 0.495$ Sv and $\overline{F_H} = 0.51$ Sv. The AMOC eventu-
ally collapses under a constant freshwater flux forcing of $\overline{F_H} = 0.51$ Sv.

The AMOC in the quasi-equilibrium simulation starts to tip around $F_H = 0.525$ Sv (van Westen et al., 2024a) and is at larger
$F_H$ values than the upper bound found from the statistical equilibria simulation (0.495 Sv $< \overline{F_H} <$ 0.51 Sv). To determine the
overshoot of quasi-equilibrium simulations, we use a reference critical value of $\overline{F_H} = 0.5$ Sv, but any other $\overline{F_H}$ value within
the interval $\overline{F_H} \in [0.495, 0.51]$ can be used as a reference (giving slightly different numerical results). Using this reference
the quasi-equilibrium AMOC overshoots by $\Delta F_H = 0.025$ Sv ($\approx$ 80 years). This overshoot is substantially smaller than the
0.2 Sv overshoot found in the FAMOUS model (Hawkins et al., 2011), but note that the freshwater flux forcing in FAMOUS
was varied by $5 \times 10^{-4}$ Sv yr$^{-1}$, about 2.5 times larger than used in the CESM. The AMOC strength of the quasi-equilibrium
remains close to the five different statistical equilibria and the time means (last 50 years) AMOC strengths of these statistical
equilibria are about 1 Sv stronger than the quasi-equilibrium (around the same freshwater flux forcing). For $F_{\mathrm{ovS}}$, on the other
hand, the quasi-equilibrium is larger and (mostly) outside the ranges of the different statistical equilibria (Figure 2b).

When we lower the freshwater flux forcing rate, we expect that the system stays closer to the statistical equilibria (Hawkins
et al., 2011). To test this, we branched off a quasi-equilibrium simulation with only half the hosing rate (i.e., $1.5 \times 10^{-4}$ Sv yr$^{-1}$)
from the end of the statistical equilibrium at $\overline{F_H} = 0.45$ Sv. This simulation was integrated for 1,050 model years, where $F_H$
varied from 0.45 Sv to 0.608 Sv (red curves in Figures 3a,b). The half-rate forcing quasi-equilibrium simulation remains (very)
close to the different statistical equilibria for both AMOC strength and $F_{\mathrm{ovS}}$. The AMOC eventually collapses around $F_H =$
0.53 Sv and there is an overshoot of $\Delta F_H = 0.03$ Sv ($\approx$ 200 years) compared to our reference critical value of $\overline{F_H} = 0.5$ Sv.
Surprisingly, the overshoot in the half-rate forcing quasi-equilibrium simulation ($\Delta F_H = 0.03$ Sv) is slightly larger than the
standard quasi-equilibrium simulation ($\Delta F_H = 0.025$ Sv).

To understand the larger overshoot under a half-rate forcing, we decompose the different AMOC feedbacks following the
procedure outlined in Vanderborght et al. (2024). The most important feedbacks are shown in Figure 4 for the half-rate forcing
simulation and for constant $\overline{F_H} = 0.51$ Sv simulation (branched from the previous statistical equilibrium of $\overline{F_H} = 0.495$ Sv).
For the latter (Figure 4b), the AMOC weakens by about 1.5 Sv during the first 100 model years. This weakening is attributed to







**Figure 2.** (a): The AMOC strength at 1,000 m and 26°N and (b): the freshwater transport by the AMOC at 34°S, $F_{\mathrm{ovS}}$, for varying freshwater flux forcing $F_H$ (i.e., the quasi-equilibrium simulation). Inset: The hosing experiment where fresh water is added to the ocean surface between 20°N – 50°N in the Atlantic Ocean ($+F_H$) and is compensated over the remaining ocean surface ($-F_H$). The statistical equilibria for various constant values of $F_H$ (i.e., $\overline{F_H}$) in the northward overturning regime are also shown (i.e., steady states), where the marker indicates the mean and the error bars show the minimum and maximum over the last 50 years of the 500-year long branched simulations. The black sections indicate the 26°N and 34°S latitudes over which the AMOC strength and $F_{\mathrm{ovS}}$ are determined, respectively. The yellow shading in the two panels indicates observed ranges for the presented quantity (Smeed et al., 2018; Arumí-Planas et al., 2024). (c – n): Similar to panels a,b, but now the entire branched simulations for different $\overline{F_H}$ values. The branches are initiated from the quasi-equilibrium simulation (blue curves) or from the end of the previous statistical equilibria (red curves).







**Figure 3.** (a & b): The AMOC strength and $F_{\text{ovS}}$ of the quasi-equilibrium simulations, one similar to Figures 2a,b, and including the simulation with varying $1.5 \times 10^{-4}$ Sv yr$^{-1}$ hosing rate (red curves). This quasi-equilibrium hosing with $1.5 \times 10^{-4}$ Sv yr$^{-1}$ was branched from the end of the statistical equilibria at $\overline{F_H} = 0.45$ Sv. (c & d): The variance in AMOC strength and $F_{\text{ovS}}$, using a sliding window of 50 years. For each 50-year window, a linear trend was removed and then the variance was determined.





the slightly larger freshwater forcing (+0.015 Sv) compared to the starting equilibrium solution at $\overline{F_H} = 0.495$ Sv. The desta-bilizing salt-advection feedback (linked to $F_{ovS}$) and surface (mainly sea-ice melt) feedback slowly grow over the following 250 years. Over the same period (model years $100 - 350$), the gyres and overturning component at 65°N partly stabilize the AMOC. The combined effect results in an AMOC weakening of only 1.5 Sv over 250 years and thereafter the AMOC fully collapses (Figure 4b). This means that the destabilizing AMOC feedbacks fully develop on a centennial time scale (under constant freshwater flux forcing).

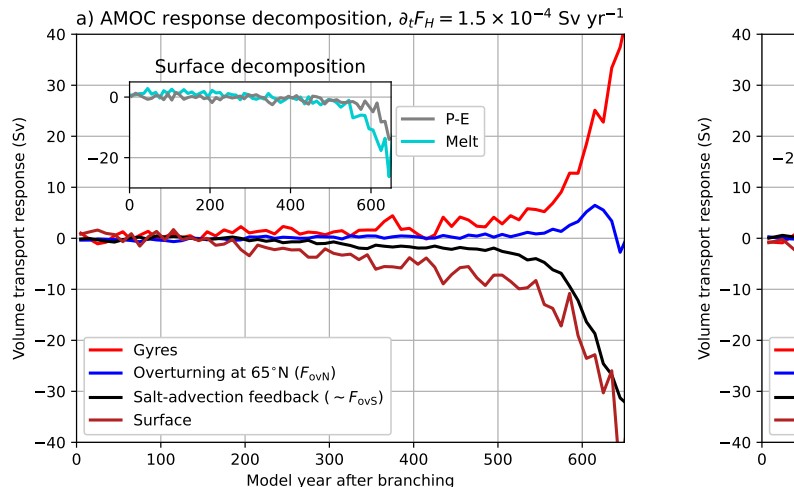

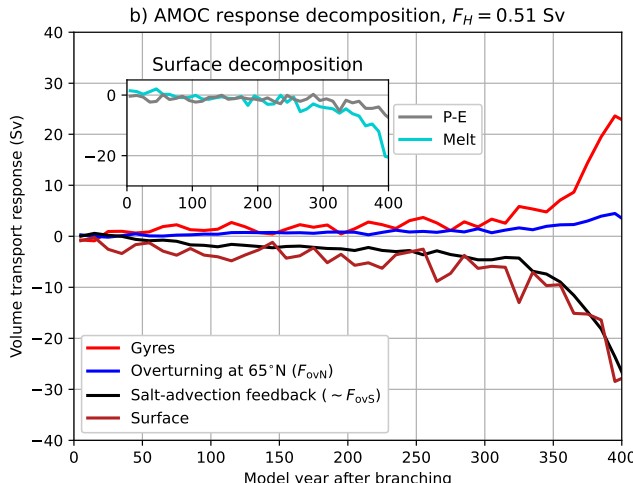

**Figure 4.** (a & b): Decomposition of the AMOC feedbacks for the (a): half-rate forcing simulation ($1.5 \times 10^{-4}$ Sv yr$^{-1}$) and the (b): $\overline{F_H} = 0.51$ Sv simulation (i.e., red curve in panel 2h). The inset shows the two surface components of Arctic sea-ice melt and precipitation minus evaporation (P-E). The time series are presented as 10-year averages (to reduce the variance). Note the different horizontal ranges between the two panels.

For the half-rate forcing simulation (Figure 4a), we find a similar centennial time scale for the destabilizing AMOC feed-backs. The AMOC feedbacks remain relatively small up to model year 350 ($F_H = 0.503$ Sv), then slowly increase in the following 200 years (model years $350 - 550$) and thereafter the AMOC fully collapses. This gradual increase of the destabiliz-ing feedbacks suggests that the critical value of forcing was crossed around $F_H = 0.503$ Sv, which is well within the interval $0.495$ Sv $< \overline{F_H} < 0.51$ Sv (or $\pm 50$ model years).

The centennial timescale ($\approx 200$ model years) over which the destabilizing feedbacks grow also gives an indication why the branched simulations from the standard quasi-equilibrium simulation show an AMOC collapse for $\overline{F_H} > 0.465$ Sv (blue curves in Figures 2c – n). The AMOC collapse starts at $F_H = 0.525$ Sv in the standard quasi-equilibrium simulation, meaning that the destabilizing feedbacks were growing during the 200 model years ($\Delta F_H = 0.06$ Sv) prior to the collapse (Vanderborght et al., 2024). This suggests that $\overline{F_H} = 0.525 - 0.06 = 0.465$ Sv is the latest statistical equilibrium which can be found when directly branching from the quasi-equilibrium simulation, which is indeed the case here (Figures 2e,k). Note, however, that additional





simulations are needed to make this more precise, but the ones we have conducted at least indicate that the destabilizing feedbacks develop on centennial timescales in a quasi-equilibrium approach.

What is important here, is that the half-rate forcing's transition to the collapsed state is twice as fast (in $F_H$ space), which is a typical characteristic of transitions near a saddle-node bifurcation (Kuehn, 2011). The duration of AMOC transitions in both quasi-equilibria and in the statistical equilibrium simulations (Figure 2) is about 100 years and the full equilibration to the collapsed AMOC state requires more than 500 years (van Westen et al., 2024a). Another characteristic of a saddle-node bifurcation is the loss of resilience (i.e., critical slow down) near the tipping point (van Westen et al., 2024b). This can be

quantified by determining the variance and (lag-1) autocorrelation of specific observables. For the AMOC strength, we find no indications of critical slow down (not shown) which is consistent with the results in van Westen et al. (2024a). There is also no increase in the variance for the AMOC strength for both the quasi-equilibria and the statistical equilibria (Figure 3c). However, for the physics-based quantity $F_{\mathrm{ovS}}$ we find indications of critical slowdown (van Westen et al., 2024a). Indeed, the $F_{\mathrm{ovS}}$ variance increases for larger $F_H$ up to the tipping event (Figure 3d). This increase in variability indicates that the AMOC

loses resilience and makes it more prone to transitions.

## 3.2   Equilibria in the E-CCM

We performed similar quasi-equilibrium and equilibrium simulations with the E-CCM, and these are presented together with the steady states from the continuation technique (cf. section 2b) in Figure 5. Note that we used slightly different freshwater flux forcing ($E_A$) values in the E-CCM than in the CESM. The continuation indicates two saddle-node bifurcation at $E_A^1 =$

0.4861 Sv (AMOC on) and at $E_A^2 = 0.1857$ Sv (AMOC off). The responses in the E-CCM are (qualitatively) comparable to that of the CESM (compare Figures 2 and 5). In the quasi-equilibrium simulation the AMOC strength is lower compared the value at the steady states, while the $F_{\mathrm{ovS}}$ values are higher. The branches from the quasi-equilibrium eventually collapse (for $\overline{E_A} = 0.477$ Sv and $\overline{E_A} = 0.486$ Sv) but a steady state can still be found for $\overline{E_A} = 0.477$ Sv, when starting from a previous steady state.

When the hosing is only applied over box n ($\xi = 0$, Figure 6a), the AMOC strength in the quasi-equilibrium ($3 \times 10^{-4}$ Sv yr$^{-1}$) undershoots the steady AMOC on state and collapses when reaching the freshwater flux forcing values of $\overline{E_A} \geq 0.474$ Sv. Note that this lower bound is found at smaller freshwater flux forcing values than at the saddle-node bifurcation of the AMOC on state ($E_A^1 = 0.4861$ Sv). The quasi-equilibrium simulation doesn't cross the basin boundary for the AMOC strength (left inset in Figure 6a) and collapses when keeping the freshwater flux flux forcing constant at $\overline{E_A} = 0.474$ Sv. When we analyse a dif-

ferent quantity, such as the salinity of box n (right inset in Figure 6a), it does cross the basin boundary. The salinity in box n is important here as it (partly) sets the AMOC strength (relation 1) and is influenced under the destabilizing salt-advection feedback. When we lower the quasi-equilibrium rate, we can approach the saddle-node bifurcation even closer before the AMOC collapses. For example, using a ten times smaller forcing rate ($3 \times 10^{-5}$ Sv yr$^{-1}$), the AMOC equilibrates to the AMOC on state when increasing $E_A$ up to 0.483 Sv and then keeping the freshwater flux forcing constant (not shown).

When we equally distribute the hosing over box n and box t ($\xi = 0.5$, Figure 6b), the saddle-node bifurcations shift to higher values of $E_A$. This quasi-equilibrium also undershoots the stable AMOC on state and close to the saddle-node bifurcation it





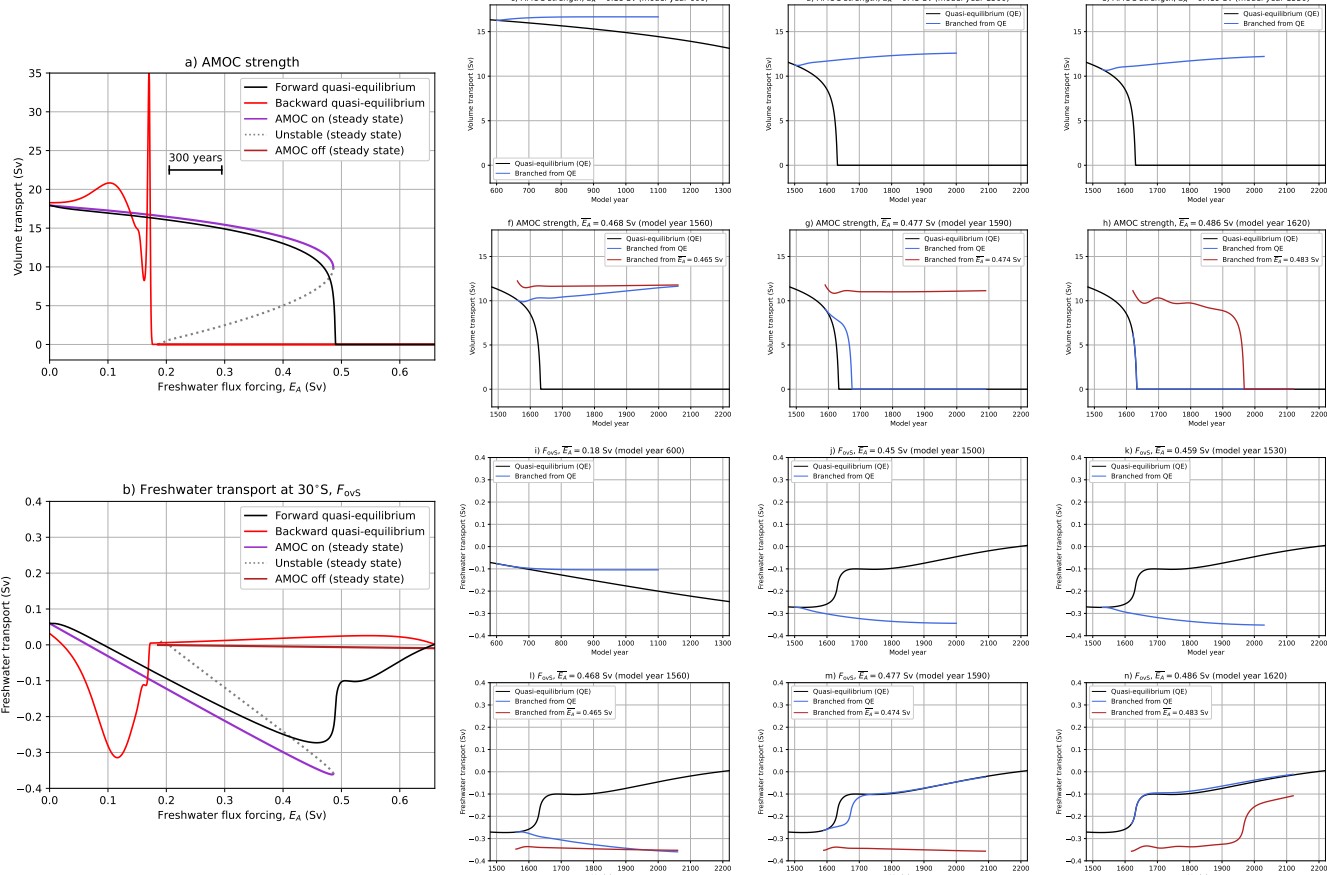

**Figure 5.** Similar to Figure 2, but now for the E-CCM. Note that in panels a and b the steady and unstable states (from the continuation) are also shown.

starts to overshoot the AMOC on state (left inset in Figure 6b). Although the quasi-equilibrium has a stronger AMOC strength than the steady AMOC on state, it still collapses when keeping the freshwater flux forcing constant at $\overline{E_A} = 0.609$ Sv. The salinity in box n crosses the basin boundary (right inset in Figure 6b), demonstrating again that AMOC strength is no good indicator for predicting an AMOC collapse. Only when the hosing is applied over box t ($\xi = 1.0$, Figure 6c), the AMOC collapses when increasing the freshwater flux forcing beyond the saddle-node bifurcation of $E_A^1 = 0.83495$ Sv. When we increase $E_A$ up to 0.8348 Sv for the quasi-equilibrium, the solution under constant forcing equilibrates to the stable AMOC on state (see insets in Figure 6c).

The undershoot of the AMOC strength can be understood from these three different cases. When a hosing perturbation is (partly) applied over box n, the AMOC strength directly reduces as the meridional salinity difference between box n and box ts increases. The largest part of the freshwater perturbation is carried away by the AMOC to box d, but a small part of the perturbation remains in box n (due to a weaker AMOC) and causes freshwater accumulation over box n. This freshwater accu-

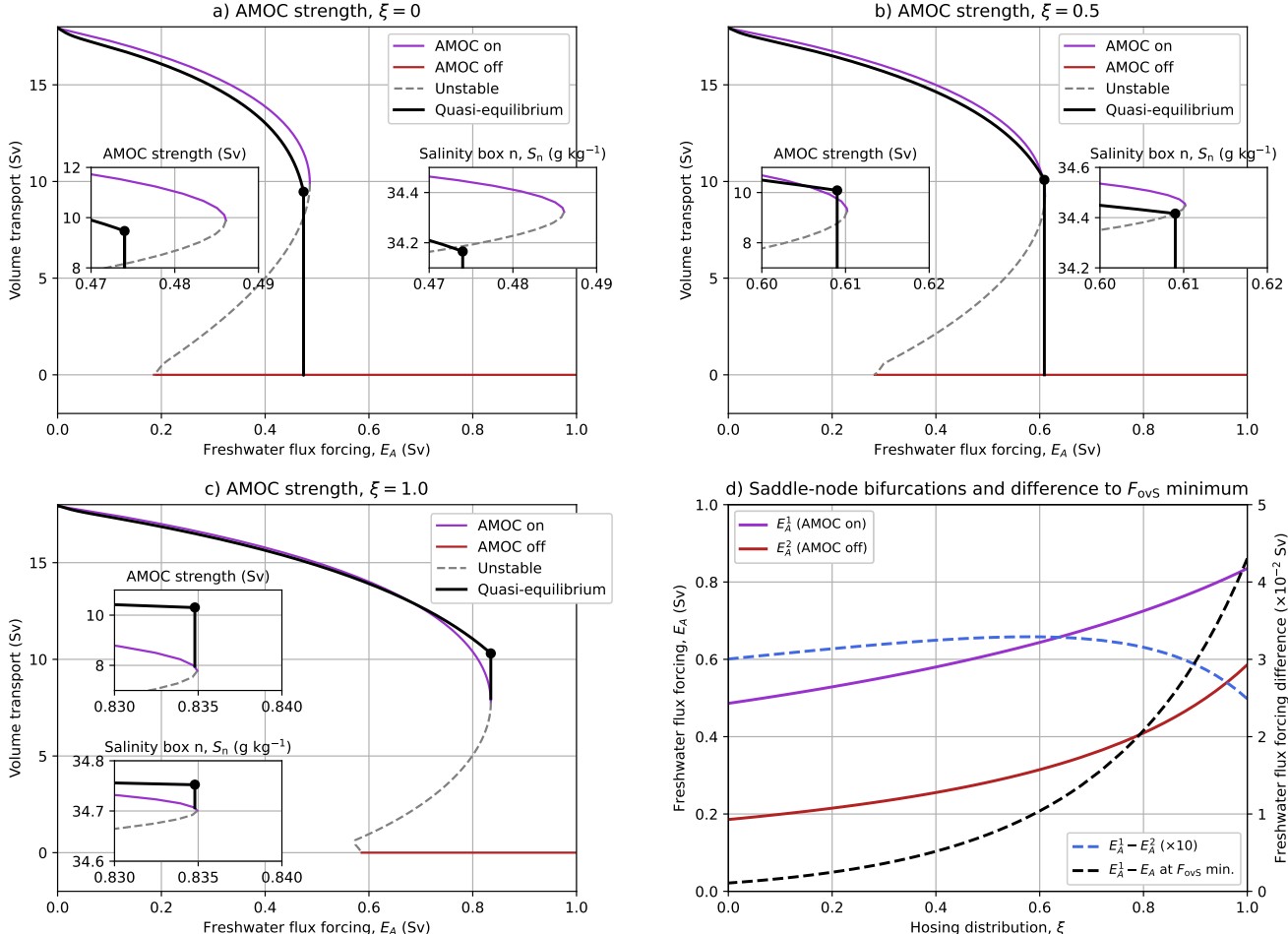

**Figure 6.** (a): The steady states for the AMOC strength and a quasi-equilibrium simulation (rate $3 \times 10^{-4}$ Sv yr$^{-1}$) for the hosing over box n ($\xi = 0$). For the quasi-equilibrium, the maximum freshwater flux forcing increases up to $E_A = 0.474$ Sv (black dot) and remains constant thereafter. The two insets show zoomed-in versions of the AMOC strength and salinity of box n near the saddle-node bifurcation. (b&c): Similar to panel a, but now for b) $\xi = 0.5$ and c) $\xi = 1$, where the maximum freshwater flux forcing increases up to $E_A = 0.609$ Sv and $E_A = 0.8348$ Sv, respectively. (d): The position of the saddle-node bifurcations of the AMOC on ($E_A^1$) and AMOC off ($E_A^2$) states (solid curves). The distance (expressed in $\Delta E_A$) between $E_A^1$ and $E_A^2$ and between the $E_A^1$ and the $F_{\mathrm{ovS}}$ minimum.





mulation results in a slightly weaker AMOC strengths compared to the steady states. Once the system has a sufficient amount of time to adjust to the imposed freshwater perturbation, the entire freshwater perturbation is redistributed over the boxes and the

AMOC strength eventually increases (e.g., blue curves in Figures 5c,d,e,f). In other words, the advective ('flushing') timescale is slower than the hosing timescale, resulting in an enhanced AMOC strength decline.

This direct AMOC weakening effect is smaller when adding (part of) the hosing over box t and there are two effects contributing to this different behaviour. First, the hosing is now distributed over the (much) larger box t than box n and making the freshwater anomalies (averaged over box t) effectively smaller. Second, only a part of the freshwater perturbations from

box t is carried by the AMOC into box n and most of it is directly carried to box d (see also Figure 1). This implies that the role of the overturning contribution in redistributing freshwater anomalies between box t and box n is getting smaller, while the (northern) gyre contribution is getting more important. These combined effects explain why the saddle-node bifurcations shift to larger $E_A$ values for increasing $\xi$ (Figure 6d). The larger gyre contribution is also reflected in a greater $\Delta E_A$ between the $E_A^1$ and $F_{\text{ovS}}$ minimum, which also modifies the hysteresis width which is measured as the distance between the two saddle-node

bifurcations (Figure 6d). In the standard quasi-equilibrium CESM simulation (rate $3 \times 10^{-4}$ Sv yr$^{-1}$), the AMOC strength is also smaller than that of the statistical equilibria. This undershooting AMOC is expected to depend on the hosing region (here 20°N – 50°N) and we expect changes in the width of the multi-stable regime when varying the hosing region (Rahmstorf, 1996; Ma et al., 2024), similar as was demonstrated here for the E-CCM when varying $\xi$.

## 4   Feedback analysis in the CESM

From Section 3.1, it is clear that the expected square root dependence of the AMOC strength (in the AMOC on state) on the increasing freshwater flux forcing near a saddle-node bifurcation, as in the Stommel (1961) model (see Appendix), can not easily be demonstrated using only a limited number of equilibrium simulations. However, it turns out that from performing a feedback analysis as in Vanderborght et al. (2024), we can (under reasonable assumptions) derive a reduced model explicitly showing the dependence of AMOC strength on $F_H$.

### 4.1   Reduced model derivation

We start from the total Atlantic (34°S to 65°N) freshwater budget as governed by (Vanderborght et al., 2024):

$$\frac{\mathrm{d}W}{\mathrm{d}t} = F_{\text{azS}} - F_{\text{azN}} + F_{\text{ovS}} - F_{\text{ovN}} + F_{\text{surf}} + F_{\text{res}}, \tag{2}$$

where $W$ is the total freshwater content. The Atlantic freshwater content can be modified through azonal (gyre) contributions (i.e., $F_{\text{azS}}$ and $F_{\text{azN}}$), overturning contributions (i.e., $F_{\text{ovS}}$ and $F_{\text{ovN}}$), surface contribution (i.e., $F_{\text{surf}}$) and residual contribution

(i.e., $F_{\text{res}}$). The quantities $F_{\text{azS}}$ and $F_{\text{ovS}}$ are evaluated at 34°S, hence indicated with subscript 'S', and we follow a similar notation for the northern boundary (65°N) by using a subscript 'N'.

Upon a freshwater perturbation, the evolution of the different contributions depends on the background state and the AMOC strength (Vanderborght et al., 2024). The AMOC strength is fairly homogeneous over the Atlantic basin (van Westen et al.,



2024a) and we assume a northward volume transport in the upper AMOC limb which we indicate here as $\Psi$; the lower AMOC limb then carries $\Psi$ southward. The average salinity content over the upper AMOC limb is indicated with $S_{\rightarrow}$, and for the salinity content over the lower AMOC limb we use $S_{\leftarrow}$. The vertical salinity difference between the upper AMOC limb and lower AMOC limb is then indicated by $S_{\rightleftarrows} = S_{\rightarrow} - S_{\leftarrow}$. Under this idealization it directly follows that:

$$F_{\mathrm{ovS}} = -\frac{S_{\rightleftarrows}}{S_0}\Psi, \tag{3}$$

where $S_0 = 35$ g kg$^{-1}$. Because the salinity transport in the lower AMOC limb is approximately adiabatic, the vertical salinity contrast at 34°S is closely related to a meridional salinity contrast between 34°S and the North Atlantic sinking region. This meridional salinity contrast is related to the AMOC strength via thermal wind balance (Butler et al., 2016). Therefore, the vertical salinity contrast scales with the AMOC strength as (Vanderborght et al., 2024):

$$\Psi = \Psi_0 + c_2\left(1 - c_1\right)\left(S_{\rightleftarrows}(0) - S_{\rightleftarrows}\right), \tag{4}$$

where $c_1$ represents the stabilizing thermal-advective feedback and $c_2$ is a scaling factor. Both $c_1$ and $c_2$ are positive constants and, for the CESM, their values are about 0.52 and 20 Sv kg g$^{-1}$ (Vanderborght et al., 2024). The terms $\Psi_0$ and $S_{\rightleftarrows}(0)$ are the AMOC strength and vertical salinity difference for $F_H = 0$ Sv (no hosing).

Under the applied hosing (indicated by $\delta F_H$ in the CESM) the value of $F_{\mathrm{surf}}$ increases and is primarily (i.e., to first order) balanced by a declining $F_{\mathrm{ovS}}$ (van Westen et al., 2024a). On the other hand, the gyres flush freshwater anomalies out of the Atlantic Ocean and stabilize the AMOC (Vanderborght et al., 2024). Sijp (2012) argued that $S_{\rightleftarrows}$ linearly scales with the integrated Atlantic freshwater content. This integrated freshwater content in turn scales with the anomalous freshwater transport by the gyres (Huisman et al., 2010), i.e.:

$$F_{\mathrm{gyre}} = F_{\mathrm{azS}} - F_{\mathrm{azN}} = -g_1 S_{\rightleftarrows} + g_2. \tag{5}$$

This linear relation is also applicable for the CESM, where $g_1 = 0.032$ Sv kg g$^{-1}$ and $g_2 = 0.49$ Sv (Figure 7a). The last contribution which we consider is the overturning component at the northern boundary, $F_{\mathrm{ovN}}$. The AMOC strength almost vanishes at the northern boundary and the expression for $F_{\mathrm{ovN}}$ is different than that of the $F_{\mathrm{ovs}}$ (relation 3). The $F_{\mathrm{ovN}}$ scales linearly with $S_{\rightleftarrows}$ and can be approximated by:

$$F_{\mathrm{ovN}} = n_1 S_{\rightleftarrows} + n_2 \tag{6}$$

with $n_1 = 0.025$ Sv kg g$^{-1}$ and $n_2 = -0.021$ Sv for the CESM as shown in Figure 7b. The contributions by the gyres and $F_{\mathrm{ovN}}$ scale linearly with increasing $S_{\rightleftarrows}$ (or decreasing $\Psi$), whereas the $F_{\mathrm{ovS}}$ has a non-linear contribution. We do not consider the residual ($F_{\mathrm{res}}$) or additional atmosphere-ocean-sea ice (e.g., sea-ice melt, Figure 4) contributions, as most conceptual (box) models only represent the overturning and gyre responses.

A perturbation in the Atlantic freshwater content (cf. (2)) around an equilibrium state then gives:

$$-\delta F_{\mathrm{ovS}} + \delta F_{\mathrm{ovN}} - \delta F_{\mathrm{gyre}} = \delta F_{\mathrm{surf}}, \tag{7}$$





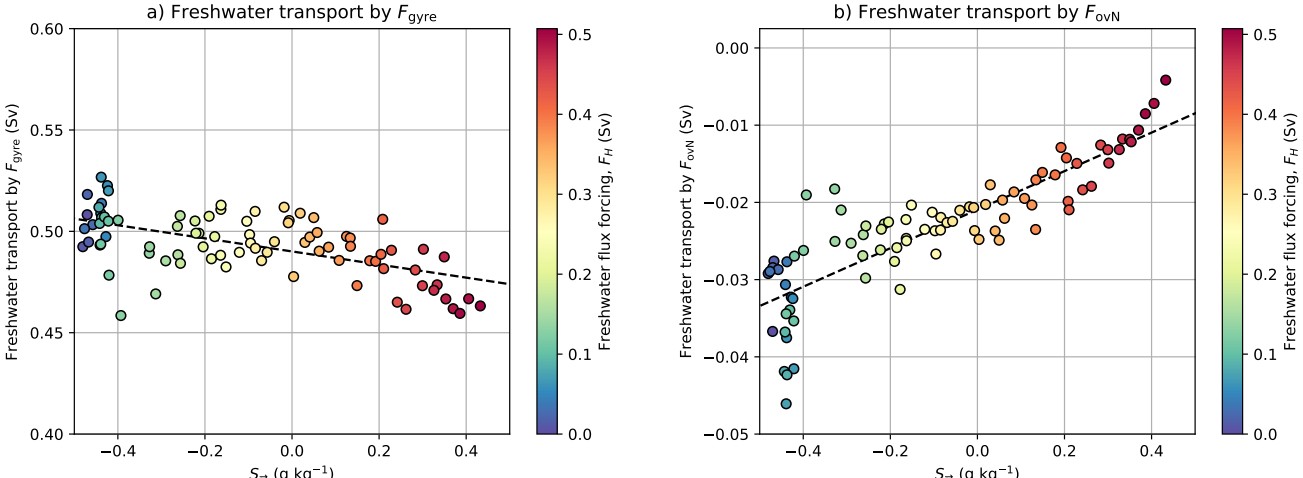

**Figure 7.** (a): The relation between $F_{\mathrm{gyre}}$ and $S_{\rightleftarrows}$, where the linear fit is determined over the 20-year averages up to model year 1,700 ($F_H = 0.51$ Sv) of the standard quasi-equilibrium simulation. (b): Similar to panel a, but now for the $F_{\mathrm{ovN}}$ and $S_{\rightleftarrows}$.

and using the expressions for $F_{\mathrm{ovS}}$, $F_{\mathrm{gyre}}$ and $F_{\mathrm{ovN}}$, this yields:

$$\Psi \delta S_{\rightleftarrows} + S_{\rightleftarrows} \delta \Psi + n_1 S_0 \delta S_{\rightleftarrows} + g_1 S_0 \delta S_{\rightleftarrows} = S_0 \delta F_H \tag{8}$$

Using the relation between $\Psi$ and $S_{\rightleftarrows}$ (from 4) we find:

$$-\frac{\Psi}{c_2 (1 - c_1)} \delta \Psi + \left( -\frac{\Psi}{c_2 (1 - c_1)} + \frac{\Psi_0}{c_2 (1 - c_1)} + S_{\rightleftarrows}(0) - \frac{(n_1 + g_1) S_0}{c_2 (1 - c_1)} \right) \delta \Psi = S_0 \delta F_H, \tag{9}$$

which can be rewritten as:

$$(-2\Psi + \Psi_0 + c_2 (1 - c_1) S_{\rightleftarrows}(0) - (n_1 + g_1) S_0) \delta \Psi = c_2 (1 - c_1) S_0 \delta F_H, \tag{10}$$

and integrating both sides gives:

$$\Psi^2 - (\Psi_0 + c_2 (1 - c_1) S_{\rightleftarrows}(0) - (n_1 + g_1) S_0) \Psi + c_2 (1 - c_1) S_0 F_H + C = 0, \tag{11}$$

with integration constant $C$. The solution with $\Psi(F_H = 0) = \Psi_0$ is:

$$\Psi(F_H) = \frac{\Psi_0}{2} + \frac{c_2 (1 - c_1) S_{\rightleftarrows}(0)}{2} - \frac{(n_1 + g_1) S_0}{2} \pm \sqrt{\left( \frac{\Psi_0 - c_2 (1 - c_1) S_{\rightleftarrows}(0) + (n_1 + g_1) S_0}{2} \right)^2 - c_2 (1 - c_1) S_0 F_H} \tag{12}$$

Rather using $S_{\rightleftarrows}(0)$, we express it as the initial $F_{\mathrm{ovS}}$ using (3), i.e., $S_{\rightleftarrows}(0) = -\frac{S_0 F_{\mathrm{ovS}}(0)}{\Psi_0}$. The final expression becomes:

$$\Psi(F_H) = \frac{\Psi_0}{2} - \frac{c_2 (1 - c_1) S_0 F_{\mathrm{ovS}}(0)}{2\Psi_0} - \frac{(n_1 + g_1) S_0}{2} \pm \sqrt{\left( \frac{\Psi_0^2 + c_2 (1 - c_1) S_0 F_{\mathrm{ovS}}(0) + (n_1 + g_1) S_0 \Psi_0}{2\Psi_0} \right)^2 - c_2 (1 - c_1) S_0 F_H}$$





265                                                                                                                                        (13)

Note that near the AMOC tipping point, the linear assumption in $F_{\mathrm{gyre}}$ and $F_{\mathrm{ovN}}$ with $S_{\rightleftarrows}$ is less accurate and $c_1$ is less constant. Moreover, other feedbacks are also important in the CESM, such as ocean-sea ice interactions (destabilizing), ocean-atmosphere fluxes (destabilizing), pycnocline deepening (stabilising), open Bering strait (stabilizing) and the effect of ocean eddies (stabilizing) (Vanderborght et al., 2024). These additional processes modify the idealized AMOC response and make

it more difficult to derive an analytical solution for the northward overturning regime, as these processes (ideally) need to be expressed as a function of $S_{\rightleftarrows}$. We stress that this idealized AMOC response under hosing should be interpreted with care and one needs to consider the appropriate feedback contributions for each (climate) model set-up. The key point is that the AMOC strength exhibits a square-root dependence on the freshwater flux forcing, leading to a saddle-node bifurcation when the dominant balance is between the applied freshwater flux forcing and the overturning component. As long as other contributions

remain sufficiently small, their effect will not change the structure (and therefore the type) of the bifurcation diagram.

For the Stommel 2-box model, we can demonstrate that a similar AMOC response holds (see Appendix). Under no freshwater flux forcing ($\eta = 0$) in this model, the salinity difference between the two boxes is zero. This constraint gives the initial AMOC strength of $\Psi_0 = k\alpha\Delta T^a$ and $F_{\mathrm{ovS}}(\eta = 0) = 0$, where $k$ is a hydraulic pumping coefficient, $\alpha$ the (dimensionless) thermal expansion coefficient, and $\Delta T^a$ the (dimensionless) atmospheric temperature difference. The northern boundary is

closed ($n_1 = 0$) and gyres are not represented ($g_1 = 0$) in the Stommel model. The oceanic temperatures in the Stommel model are fixed (under steady state assumption), and in this case $c_1 = 0$. Relation (13) for the Stommel model reduces to:

$$\Psi(F_H) = \frac{k\alpha\Delta T^a}{2} \pm \sqrt{\left(\frac{k\alpha\Delta T^a}{2}\right)^2 - c_2 S_0 F_H} \tag{14}$$

and is similar to relation A9, apart from some scaling coefficients.

### 4.2 Application of the reduced model

Using the reduced model, the critical value of $F_H$ for an AMOC collapse in the CESM can be estimated by assuming that the freshwater flux forcing is (in its first order) balanced by the overturning and azonal (gyre) components, which is the case for the CESM (van Westen et al., 2024a). The critical freshwater flux forcing is obtained by setting the terms under the square root in equation (13) equal to zero. Solving this yields:

$$F_H^c = \frac{1}{c_2\left(1 - c_1\right)S_0} \left(\frac{\Psi_0^2 + c_2\left(1 - c_1\right)S_0 F_{\mathrm{ovS}}(0) + (n_1 + g_1)S_0\Psi_0}{2\Psi_0}\right)^2. \tag{15}$$

The $F_H^c$ is dependent on the initial AMOC strength and initial $F_{\mathrm{ovS}}$ value. In the CESM, the Atlantic Ocean surface area outside 20°N – 50°N receives a negative freshwater flux as part of the global compensation (see inset Figure 2a). This makes the applied hosing 86% effective when considering the total Atlantic Ocean surface area (34°S – 65°N) and $F_H^c$ needs to be adjusted by a factor $\frac{1}{0.86}$. The time-means (first 50 model years) in the CESM quasi-equilibrium simulation are $\Psi_0 = 16$ Sv and $F_{\mathrm{ovS}}(0) = 0.22$ Sv, which give: $F_H^c = \frac{1}{0.86}0.38 = 0.44$ Sv (Figures 8a,b). When using the maximum and minimum values





**Figure 8.** (a&b): The AMOC and $F_{\mathrm{ovS}}$ responses of the reduced model under the freshwater flux forcing (cf. equations (13) and (3), respectively), where the solid curves indicate the steady AMOC on state and dotted curves the unstable branch. The initial values for both the AMOC strength and $F_{\mathrm{ovS}}$ were obtained from the first 50 model years of the quasi-equilibrium. The AMOC strength values are 16.0 Sv (mean), 17.8 Sv (maximum) and 14.3 Sv (minimum), and $F_{\mathrm{ovS}}$ values are 0.22 Sv (mean), 0.24 Sv (maximum) and 0.20 Sv (minimum). For estimates from observations we used 17 Sv (Smeed et al., 2018) and $-0.15$ Sv (Arumí-Planas et al., 2024) for the AMOC strength and $F_{\mathrm{ovS}}$, respectively. (c): The critical freshwater flux forcing ($F_H^{\mathrm{c}}$) for varying initial AMOC strength and initial $F_{\mathrm{ovS}}$. The ranges for the CESM (first 50 model years of quasi-equilibrium) and observations are indicated. The critical freshwater flux forcing was not determined for relatively weak AMOC strengths ($< 5$ Sv). (d): Values of $F_H^{\mathrm{c}}$ (solid curves) and difference to $F_{\mathrm{ovS}}$ minimum (dashed curves) for varying gyre sensitivity ($g_1$) and two cases for the northern overturning sensitivity ($n_1$), using the time-mean (first 50 model years) AMOC strength and $F_{\mathrm{ovS}}$. The standard CESM values are $g_1 = 0.032$ Sv kg g$^{-1}$ (blue dotted line) and $n_1 = 0.025$ Sv kg g$^{-1}$ (black curves). For all panels, we consider the hosing over $20°\mathrm{N} - 50°\mathrm{N}$ (with global surface compensation), making the applied hosing 86% effective (see main text).



(over the first 50 model years) for AMOC strength and $F_{\text{ovS}}$, we find $F_H^{\text{c}} = \frac{1}{0.86}0.44 = 0.52$ Sv and $F_H^{\text{c}} = \frac{1}{0.86}0.33 = 0.38$ Sv, respectively (Figures 8a,b).

The $F_H^{\text{c}}$ determined from the reduced model is somewhat smaller (0.06 Sv for the mean) than that estimated by the CESM statistical steady state computations ($\overline{F_H} = 0.5$ Sv). By increasing the gyre (or northern overturning) responses, we can reduce this difference (Figure 8d). The gyre contributions also control the distance between $F_H^{\text{c}}$ and value of $F_H$ at the $F_{\text{ovS}}$ minimum

(Dijkstra, 2007; Huisman et al., 2010; Dijkstra and van Westen, 2024). For the reduced model and with standard values of the parameters $n_1$ and $g_1$, this difference is about $\Delta F_H = 0.34 \times 10^{-2}$ Sv (Figure 8d), and decreasing with smaller $g_1$ (or $n_1$).

The actual $F_{\text{ovS}}$ minimum in CESM is found for the statistical equilibrium of $\overline{F_H} = 0.48$ Sv (Figure 9a), whereas the $F_{\text{ovS}}$ minimum in the quasi-equilibrium was found around $F_H = 0.52$ Sv (van Westen et al., 2024a). There is, however, substantial overlap in the statistical properties of the four statistical equilibria closest to the tipping point. Following van Westen et al.

(2024a), we use cubic splines that interpolate cubic polynomials between so-called knots, for these knots we use the $F_{\text{ovS}}$ values from these four statistical equilibria. For each of the four statistical equilibria (i.e., the knots), we draw one random $F_{\text{ovS}}$ value (50 years in total) and these are used to generate the cubic splines with two different boundary conditions (i.e., not-a-knot and natural). Ten random cubic splines are displayed in Figure 9a (thin curves) and the mean over 100,000 random cubic splines (thick curve) goes through the time means of the statistical equilibria. The $F_{\text{ovS}}$ minimum is found for $\overline{F_H} \leq 0.487$ Sv in 66%

of the cases (bars in Figure 9b), with the $F_{\text{ovS}}$ minimum at a mean value of $\overline{F_H} = 0.481$ Sv (from the 100,000 realisations). The $F_{\text{ovS}}$ minimum estimated from the cubic splines is frequently found at $\overline{F_H} = 0.495$ Sv (curves in Figure 9b), which is attributed to the random sampling such that the knot at $\overline{F_H} = 0.495$ Sv has the lowest $F_{\text{ovS}}$ value of the four knots. The cubic spline mean $F_{\text{ovS}}$ minimum is found $\Delta F_H = 0.014$ to 0.029 Sv before the upper bound of the multi-stable regime. A similar freshwater flux forcing difference is found in a fully-implicit global ocean model (Dijkstra and van Westen, 2024), where it

was shown that the $F_{\text{ovS}}$ minimum is connected to a saddle-node bifurcation.

Using the reduced model, one can make a rough estimate of the critical freshwater flux forcing needed to collapse the present-day AMOC, under the unrealistic assumptions that the CESM and the freshwater hosing between $20°\text{N} - 50°\text{N}$ (with global compensation) mimic global climate change. In this case, using a present-day AMOC strength of 17 Sv (Smeed et al., 2018) with a $F_{\text{ovS}}$ of $-0.15$ Sv (Arumí-Planas et al., 2024), we find $F_H^{\text{c}} = \frac{1}{0.86}0.19 = 0.22$ Sv (Figure 8). As the CESM

slightly underestimates the $F_H^{\text{c}}$ from the reduced model, at least a freshwater flux forcing of about 0.25 Sv would be needed, boiling down to 30 times the present-day melt rate of the Greenland Ice Sheet (Sasgen et al., 2020). The AMOC responses under varying $F_H$ (i.e., $\frac{\partial \text{AMOC}}{\partial F_H}$) are more sensitive in observations than in the CESM, because the present-day AMOC has a negative $F_{\text{ovS}}$ (Arumí-Planas et al., 2024) and hence in the destabilising regime, while the initial $F_{\text{ovS}}$ is positive in the pre-industrial CESM. Note that the present-day AMOC is not only forced by enhanced Greenland Ice Sheet melt, but also by

higher atmospheric temperatures (van Westen et al., 2024c) and so the above estimate of $F_H^{\text{c}}$ may not be very useful under transient climate change conditions.





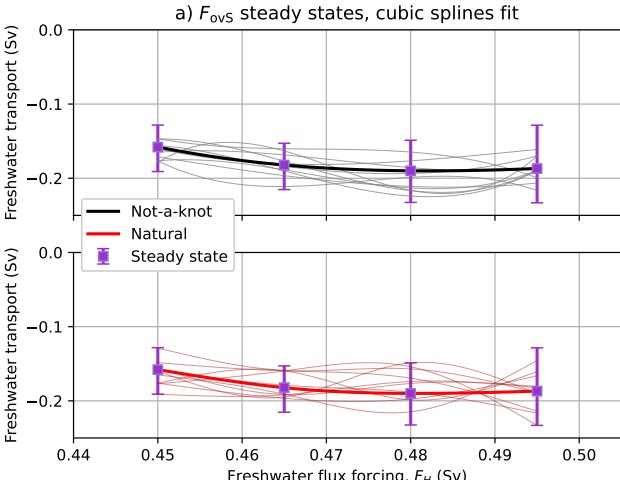
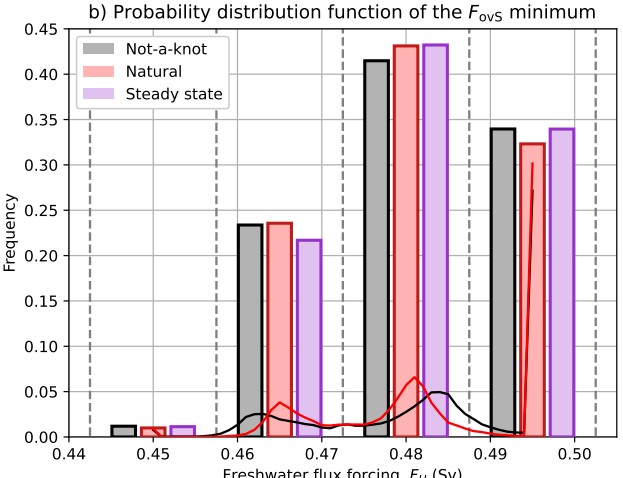

**Figure 9.** (a): Cubic splines fits (thin curves) using random $F_{ovS}$ values from the four statistical equilibria. The mean over 100,000 random cubic splines are shown by the thick curves. We use the not-a-knot boundary condition (upper panel) and the natural boundary condition (lower panel). (b): The probability distribution function (PDF) of the $F_{ovS}$ minimum using cubic splines and the expected PDF from the statistical equilibria are indicated by the bars (grouped by 0.015 Sv). For the cubic splines we also determined the PDFs with a finer resolution of 0.001 Sv (curves). For each PDF, we generated 100,000 independent sets of $F_{ovS}$ values from the four statistical equilibria.

## 5  Transient AMOC behavior under climate change

The existence of a saddle-node bifurcation in the E-CCM helps to understand how AMOC stability in CESM is influenced under climate change. Changes in the background climate conditions can be interpreted as a shift in the position of the saddle-node bifurcation. This can already be demonstrated in the Stommel model where the saddle-node bifurcation shifts to lower freshwater flux forcing values under a smaller atmospheric temperature gradient (Figure A2).

We first analyse the CESM simulations under the Hist/RCP4.5 and Hist/RCP8.5 scenarios. The AMOC collapses in three out of the four CESM simulation under climate change (Figures 10a,b). The simulation under the higher freshwater flux forcing of $\overline{F_H} = 0.45$ Sv are closer to the tipping point (under PI conditions) and hence are more prone to undergo transitions, which is indeed the case. For $\overline{F_H} = 0.18$ Sv, only the Hist/RCP8.5 scenario shows an AMOC collapse while in the Hist/RCP4.5 scenario the AMOC eventually recovers. In the latter scenario, the AMOC shows distinct centennial variability and this is associated with the typical overturning time scale (Winton and Sarachik, 1993).

The imposed transient climate change forcing induces above-averaged surface temperature trends (compared to the global mean) at the higher latitudes (i.e., polar amplification, Figures 10c,d). This temperature response reduces the meridional (equator-to-pole) temperature gradient and may influence the multi-stable AMOC regime, as is the case for the Stommel model (Figure A2). We can test this in the E-CCM by reducing the atmospheric meridional temperature gradient by imposing





**Figure 10.** (a&b): The AMOC strength at 1,000 m and 26°N under the different climate change scenarios, the yellow shading indicates observed ranges (Smeed et al., 2018). (c&d): The zonally-averaged (2-meter) surface temperature trend (model year 2000 – 2100) under the different climate change scenarios. The globally-averaged temperature trend is indicated by the dashed lines.



a (positive) atmospheric temperature anomaly ($\Delta T_n^a$) over box n (and also over atmospheric box s as they are coupled (van Westen et al., 2024b)). We keep the atmospheric temperatures the same for boxes t and ts to limit the degrees of freedom.

The steady states (with $\xi = 0$) for the reference case ($\Delta T_n^a = 0°C$) and climate change case ($\Delta T_n^a = 5°C$) are shown in
Figure 11a. Both saddle-node bifurcations shift to lower $E_A$ values and the hysteresis width decreases from 0.30 Sv (reference) to 0.22 Sv (climate change). This shift can be understood from the smaller meridional density difference between box n and box ts (equation (1)) due to higher temperatures and this requires a smaller freshwater flux forcing to reach the critical AMOC strength corresponding to the tipping point. The reduced meridional temperature gradient also weakens the AMOC on strength by a few Sv when comparing the two cases. The shift of the upper saddle-node to lower $E_A$ values indicates that the AMOC
on state loses stability under climate change.

To study the transient climate change forcing in the E-CCM, we linearly increase $T_n^a$ by 1°C per century up to model year 500 and then keep the temperature anomaly constant at $\Delta T_n^a = 5°C$. The AMOC strength (black curve in Figure 11b) under climate change is shown for constant $\overline{E_A} = 0.335$ Sv, a similar set-up as in the CESM. For each temperature anomaly $\Delta T_n^a$ we determined the steady states (with an accuracy of 0.1°C) and the values for the AMOC on, unstable branch and AMOC
off states for $\overline{E_A} = 0.335$ Sv are also shown in Figure 11b. These steady states represent the 'frozen' bifurcation diagrams for a given temperature anomaly (insets in Figure 11b). The transient AMOC is clearly deviating from the AMOC on state. Up to model year 500, the AMOC gradually weakens and after a few oscillations eventually collapses in model year 900. These oscillations are related to a (sub-critical) Hopf bifurcation close to the saddle-node bifurcation. There is no AMOC collapse for $\overline{E_A} < 0.335$ Sv (Figure 11c) but there is always an AMOC collapse for $\overline{E_A} > 0.342$ Sv (and $\Delta T_n^a = 5°C$) as there are no
stable AMOC on states at larger $E_A$ values.

For a slightly lower freshwater flux forcing of $\overline{E_A} = 0.33$ Sv, the AMOC shows oscillatory behavior when the climate change forcing remains constant after model year 500. After model year 500, the AMOC remains stable and starts to recover to its modified AMOC on state under climate change. When we increase the $T_n^a$ temperature trend, this oscillatory behavior increases (Figure 11d). For a temperature trend of 11.85 °C per century (inset in Figure 11d), the AMOC strength (and other quantities)
crosses the basin boundary between model years 43 and 87 and the AMOC displays oscillatory behavior. These oscillations decrease in amplitude after model year 800 and then the AMOC recovers. For larger temperature trends than 11.85 °C per century the AMOC eventually collapses. The greatest AMOC weakening is found for relatively large temperature trends. It is possible to collapse the AMOC for $\overline{E_A} < 0.33$ Sv and this requires even larger climate change anomalies ($\Delta T_n^a > 5°C$). Both the temperature anomaly and rate-induced effects likely play a role here in destabilizing the AMOC, but this has not further
been investigated.

## 6 Summary and Discussion

The Community Earth System Model (CESM) as used here (version 1.0.5) is an extremely high-dimensional dynamical system, representing the interaction of the ocean, atmosphere, land and sea-ice processes. In a pre-industrial configuration, the AMOC





**Figure 11.** (a): The steady states for the AMOC strength for the standard set-up (solid curves) and under climate change (dashed curves). (b): The AMOC strength under transient climate change and $\overline{E_A} = 0.335$ Sv, where $\Delta T_n^a$ linearly increases up to $5°$C up to model year 500 (trend of $1°$C per century) and then remains constant. The steady states at $\overline{E_A} = 0.335$ Sv for each climate change anomaly (with an accuracy of $0.1°$C) are also displayed. The insets show the steady states and the transient AMOC state (black dot) at $\Delta T_n^a = 2°$C (model year 200) and $\Delta T_n^a = 4°$C (model year 400). (c): Similar to panel b, but now for different values of $\overline{E_A}$. (d): The transient AMOC strength under climate change and $\overline{E_A} = 0.33$ Sv, but now for varying temperature trends in $\Delta T_n^a$. The inset shows the transient AMOC strength for a temperature trend of $11.85°$C per century.



collapses under a quasi-equilibrium input of freshwater in the $20°\mathrm{N} - 50°\mathrm{N}$ region, with surface freshwater compensation over

the rest of the global domain (van Westen et al., 2024a).

In this paper, we have provided arguments for the case that, as in ocean-climate models lower in the model hierarchy (box models (Cessi, 1994) and fully-implicit ocean models (Dijkstra, 2007)), the AMOC collapse behavior in CESM is caused by the presence of a saddle-node bifurcation in the high-dimensional dynamical system. While one indeed would expect such a bifurcation in a deterministic dynamical system when varying a single parameter (where the saddle-node and the Hopf

bifurcation are the only two generic codimension-1 bifurcations), this is far from trivial in the CESM. The ocean component of the CESM is much more complicated with several interacting positive and negative feedbacks (Vanderborght et al., 2024) and which is forced by a rapidly varying atmosphere. So attractors of the CESM are expected to have a quite complicated geometrical structure and transitions between those (such as between the AMOC on state and AMOC off state) could in principle be much more complicated than the traditional saddle-node bifurcation picture as suggested by conceptual models

(Dijkstra, 2024).

For a saddle-node, one would have to demonstrate a square root dependence of the AMOC strength on the freshwater forcing near the collapse point. This is not feasible for the CESM due to its strong internal variability and hence our case is build using three more indirect arguments. The first argument is that in the CESM, there is a strict critical boundary of existence of the statistical steady 'AMOC on' state. We showed this by subsequent near-equilibrium computations near the collapse point in

the quasi-equilibrium simulation, similar to the approach in Hawkins et al. (2011). Such a strict boundary is characteristic of a saddle-node bifurcation as shown for the E-CCM model, with parameters somehow tuned to the CESM. Second argument is based on the CESM results with a slower freshwater forcing rate. Here, we show that the AMOC collapse precisely follows the behaviour (Ritchie et al., 2021) one would expect near a saddle-node bifurcation, i.e., with a steeper transition (in $F_H$ space) than for the standard forcing rate. The third, and probably strongest, argument relies on the assumption that overturning

freshwater transport predominately compensates any freshwater flux forcing, which holds approximately for the CESM (van Westen et al., 2024a). In this case, one can show that the AMOC strength has a square-root dependence with the freshwater forcing using a reduced model (cf. section 4).

To these arguments, we can add the support from early warning indicators as found for the CESM (van Westen et al., 2024a). A characteristic property of saddle-node bifurcations is the loss of resilience (i.e., critical slowdown) near the tipping point,

measured by the increase in variance and autocorrelation (van Westen et al., 2024b). Although these early warning indicators based on the AMOC strength were not giving any critical slowdown, optimal regions for early warning signal detection were found near $34°\mathrm{S}$ (Smolders et al., 2024). The results presented here (cf. Figure 3) show an increase in the $F_{\mathrm{ovS}}$ variance close to the tipping point. This increase in variability indicates that the AMOC loses resilience and making it more prone to transitions, characteristic of approaching a saddle-node bifurcation (van Westen et al., 2024b).

The implications of this result are substantial. First of all, it shows that, for the AMOC tipping problem, conceptual models that capture only the dominant feedbacks are useful (Dijkstra, 2024). For example, in the E-CCM only the salt-advection feedback and gyre feedback are captured which are also dominant in CESM and hence it is relatively easy to tune the behavior of the E-CCM to the CESM. Similarly, Wood et al. (2019) tuned a box model (only representing the salt-advection feedback) to the





FAMOUS (Hawkins et al., 2011) where likely due to its low resolution the gyre feedback is relatively weak. Sensitivity studies
in the conceptual model can then be used to design useful simulations in the complex model and also physical explanations
can be sought in the reduced model. Second, if the multi-stable regime of the AMOC is bounded by saddle-node bifurcations,
then the effect of model biases can be studied in terms of shifts of the saddle-node bifurcations. In fully-implicit ocean models,
it was recently shown that a bias in Indian Ocean precipitation leads to a right shift (i.e., to higher Atlantic freshwater flux
forcing strengths) of the bifurcation diagram (Dijkstra and van Westen, 2024). Our reduced model (cf. Section 4b) also shows
that positive freshwater transport biases at 34°S make the AMOC more stable under hosing. If indeed a saddle-node bifurcation
is present in all global climate models (GCMs), this would indicate that GCMs having such a bias (van Westen and Dijkstra,
2024) would be too stable.

So far, the saddle-node bifurcation was discussed only in the case of an AMOC collapse when changing the freshwater flux
forcing. However, under climate change mainly the heat flux forcing will change and not in a quasi-equilibrium way. Also
in this case, we have shown that the existence of the saddle-node bifurcation is an important aspect to explain the transient
behavior of the CESM. Climate change modifies the atmospheric meridional temperature gradient and shifts the saddle-node
bifurcation to lower freshwater flux forcings, making the 'AMOC on' state less resilient. This was shown in greater detail by
the idealized results of the E-CCM, the collapse behavior can be viewed as crossing a moving saddle-node bifurcation in time
(Ritchie et al., 2021). Note that the E-CCM is limited in representing other (non-linear) climate change feedbacks, such as
enhanced evaporation (due to higher temperatures) which could partly stabilize the AMOC (van Westen et al., 2024c).

Finally, as the phase space of the CESM is so high-dimensional, why would a saddle-node bifurcation appear in such a
model (as there are many instabilities)? This result can be possibly explained by looking at the Lorenz84-Stommel1961 model
or the PlaSim sea-ice model (Tantet et al., 2018), which both display chaotic behavior, but also show a large-scale transition
under variation of one parameter. Here, the chaotic behavior is only in the atmosphere component and the large-scale transition
dynamics is governed only by the slow component, which is then noise-forced. While in the total phase space, this may be
a crisis bifurcation, in the reduced phase space of the slow component, this would appear then as a saddle-node bifurcation.
However, more work is needed to make this more precise.

*Code and data availability.* All processed model output and Python scripts to generate the results are available at:
https://doi.org/10.5281/zenodo.14510337

**Appendix A: The Analytical Solutions of the Stommel Box Model**

The Stommel 2-box model (Stommel, 1961) consists of two well-mixed boxes (equal volume) and the boxes exchange water
mass properties over time (Figure A1). The circulation strength, $\psi$, is set by the density difference between the high-latitude
$(T_1, S_1)$ and equatorial box $(T_2, S_2)$:

$$\psi = k(\rho_1 - \rho_2) \tag{A1}$$





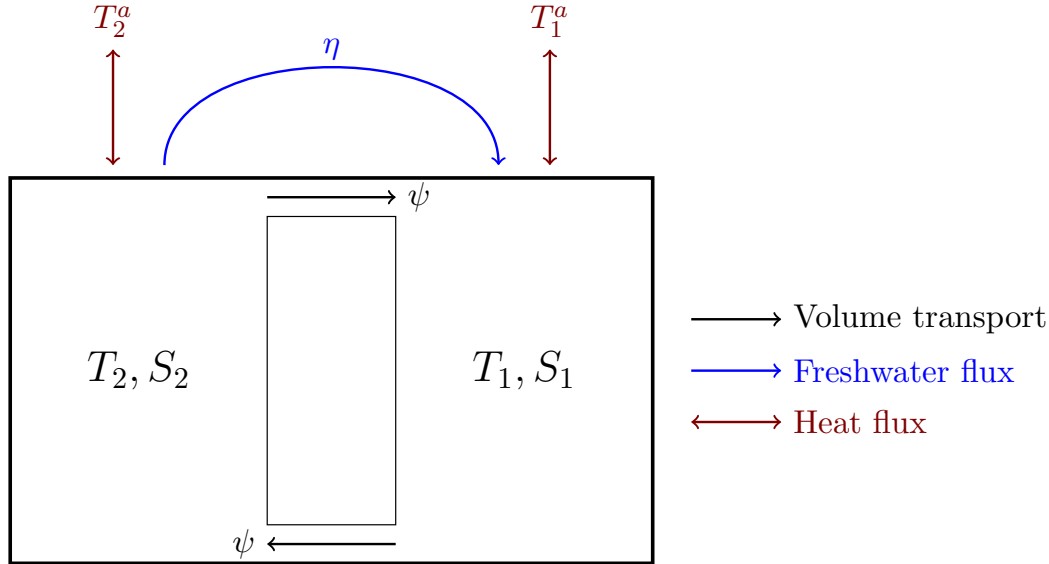

**Figure A1.** Schematic representation of the Stommel 2-box model in its northward overturning state with AMOC strength $\psi$. The blue and brown arrows are freshwater and heat fluxes, respectively. The hosing is directed from the equatorial box (with $T_2$, $S_2$) to the high-latitude box (with $T_1$, $S_1$).

where $k$ is a hydraulic pumping constant. A linear equation of state ($\rho = \rho_0 - \alpha(T - T_0) + \beta(S - S_0)$) yields:

$$\psi = k(\alpha\Delta T - \beta\Delta S) \tag{A2}$$

where $\Delta T = T_2 - T_1$ and $\Delta S = S_2 - S_1$. The governing (dimensionless) differential equation for the Stommel model are then given by:

$$\frac{\mathrm{d}T_1}{\mathrm{d}t} = |\psi|\Delta T + \lambda_T(T_1^a - T_1) \tag{A3}$$

$$\frac{\mathrm{d}T_2}{\mathrm{d}t} = -|\psi|\Delta T + \lambda_T(T_2^a - T_2) \tag{A4}$$

$$\frac{\mathrm{d}S_1}{\mathrm{d}t} = |\psi|\Delta S - \eta \tag{A5}$$

$$\frac{\mathrm{d}S_2}{\mathrm{d}t} = -|\psi|\Delta S + \eta \tag{A6}$$

In these relations $\lambda_T$ is the thermal exchange coefficient with the overhead atmosphere, the atmospheric temperatures are fixed.

Under the assumption that the thermal exchange with the atmosphere is much faster than the thermal exchange between the boxes ($\psi\Delta T \ll \lambda_T(T_i^a - T_i)$, with $i = 1, 2$), the steady state for the temperatures has $T_1 = T_1^a$ and $T_2 = T_2^a$. Using this steady state assumption, the time-evolution equation of the circulation strength (from A2 and A3 – A6) reduces to:

$$\frac{\mathrm{d}\psi}{\mathrm{d}t} = -k\beta\frac{\mathrm{d}\Delta S}{\mathrm{d}t} = -k\beta\left(\frac{\mathrm{d}S_2}{\mathrm{d}t} - \frac{\mathrm{d}S_1}{\mathrm{d}t}\right) = 2k\beta(|\psi|\Delta S - \eta) \tag{A7}$$





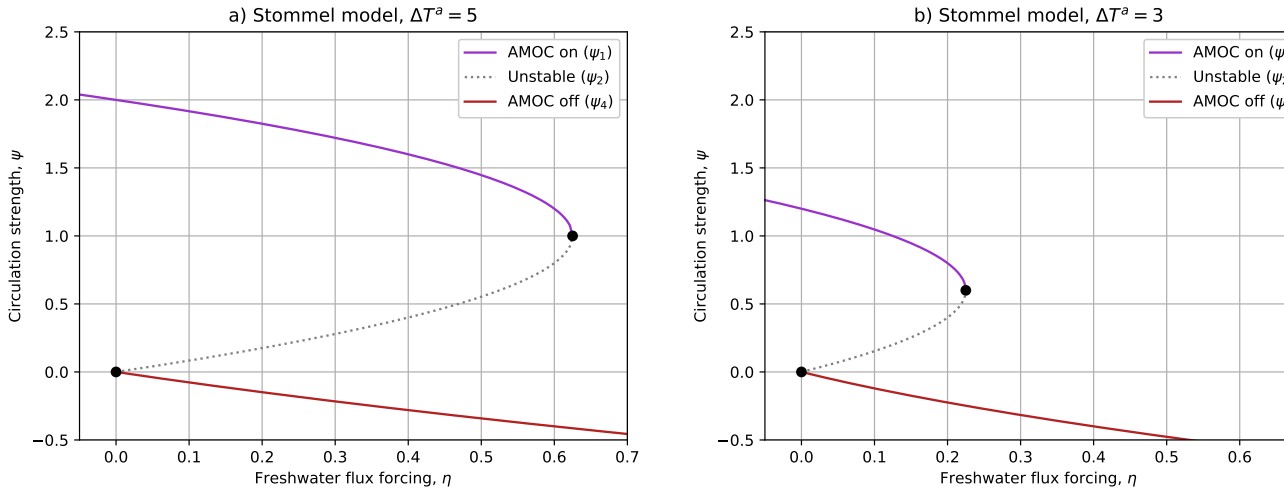

**Figure A2.** Bifurcation diagram for the Stommel 2-box model, where the black dots indicate saddle-node bifurcations. The atmospheric temperature differences are (a): $\Delta T^a = 5$ and (b): $\Delta T^a = 3$. For the other dimensionless coefficients, we used: $\alpha = 2 \times 10^{-4}, \beta = 8 \times 10^{-4}$ and $k = 2 \times 10^3$.

where the temperature contribution vanishes as the atmospheric temperatures are constant ($\frac{\mathrm{d}\Delta T}{\mathrm{d}t} = \frac{\mathrm{d}\Delta T^a}{\mathrm{d}t} = 0$). The final step
is to substitute $\Delta S = \frac{k\alpha\Delta T^a - \psi}{k\beta}$ from (A2) to obtain:

$$\frac{\mathrm{d}\psi}{\mathrm{d}t} = -2|\psi|\psi + 2k\alpha\Delta T^a|\psi| - 2k\beta\eta \tag{A8}$$

The steady states ($\frac{\mathrm{d}\psi}{\mathrm{d}t} = 0$) with northward overturning ($\psi > 0$) are given by:

$$\psi_{1,2} = \frac{k\alpha\Delta T^a}{2} \pm \sqrt{\left(\frac{k\alpha\Delta T^a}{2}\right)^2 - k\beta\eta} \tag{A9}$$

For the reversed circulation ($\psi < 0$), these are:

$$\psi_{3,4} = \frac{k\alpha\Delta T^a}{2} \pm \sqrt{\left(\frac{k\alpha\Delta T^a}{2}\right)^2 + k\beta\eta} \tag{A10}$$

but note that $\psi_3$ has to be rejected since $\psi_3 \not< 0$. The stable AMOC on state is given by $\psi_1$, the stable AMOC off state by $\psi_4$, and the unstable state by $\psi_2$. The (dimensionless) solutions for two different atmospheric temperature differences are shown in Figure A2.

*Author contributions.* R.M.v.W., E.Y. and H.A.D. conceived the idea for this study. R.M.v.W. conducted the analysis and prepared all figures,
E.Y. contributed to the AMOC feedback decomposition. All authors were actively involved in the interpretation of the analysis results and the writing process.



*Competing interests.* The authors declare no conflict of interest.

*Acknowledgements.* We thank Michael Kliphuis (IMAU, UU) for performing the additional CESM simulations. The model simulations and the analysis of all the model output was conducted on the Dutch National Supercomputer Snellius within NWO-SURF project 2024.013.
R.M.v.W., E.V. and H.A.D. are funded by the European Research Council through the ERC-AdG project TAOC (PI: Dijkstra, project 101055096).



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
