# Peer review of "A Saddle-Node Bifurcation is Causing the AMOC Collapse in the Community Earth System Model"

_EGUsphere, 2025_

## Author Comment (AC1)

**MS-No.:** egusphere-2025-14

**Version:** Revision

**Title:** A Saddle-Node Bifurcation is Causing the AMOC Collapse in the Community Earth System Model

**Author(s):** René M. van Westen, Elian Vanderborght, Henk A. Dijkstra

**Point-by-point reply to reviewer**

August 31, 2025

We thank the reviewer for their careful reading and for the useful comments on the manuscript.

*General Comments: This paper investigates the mechanisms behind Atlantic Meridional Overturning Circulation (AMOC) collapse in the Community Earth System Model (CESM). The paper aims to demonstrate that the classical picture of a saddle-node bifurcation, as exhibited by box models, also holds for the AMOC collapse in the CESM. This is done by analyzing pre-industrial CESM simulations and comparing them with a conceptual model (E-CCM). The authors use physical arguments to demonstrate that the complex behavior of AMOC in the CESM can be approximated by a reduced-order model, in which a saddle-node bifurcation drives the AMOC collapse.*

*These results underscore the utility of idealized AMOC models and may help evaluate the effect of model biases on the AMOC stability landscape and for understanding AMOC responses under various climate change projections. The careful experimental design and related analyses, for simulations in CESM and E-CCM, are commendable. I think the paper may be suitable for publication after the following issues are satisfactorily addressed.*

Specific Comments

1. *The paper needs a discussion of the normal form of a saddle-node bifurcation and its properties (this could be in the appendix), given that the central aim of the work is to establish that the AMOC, as represented in CESM, has a saddle-node bifurcation. The strongest evidence for this*

*is the square root dependence of the AMOC strength on the freshwater hosing in the reduced-order model developed from physical arguments. However, the paper does not explain why one should expect to see this square-root dependence.*

**Author's reply:**

We agree that a better introduction of the saddle-node bifurcation and the associated square-root dependence is needed in the manuscript. We will include a new appendix (now appendix A) in the paper and refer to that in the introduction.

**Changes in manuscript:**

We will revise the text accordingly.

2. *Lines 391-394: Is the second argument in support of the existence of a saddle-node bifurcation in CESM qualitative? Does the 'two-times' faster transition in the 'half-rate' hosing experiment have any quantitative significance? Isn't this related to the inertia associated with the system? Lines 150-151 claim that this is a typical characteristic of transitions near saddle-node bifurcations. Could you explain this in more detail? I was unable to verify this claim by going through Kuehn 2011. In summary, I think the authors need to precisely clarify how this observation constitutes evidence for the existence of a saddle-node bifurcation.*

**Author's reply:**

The two times faster transition in forcing space for the half rate experiment is a characteristic of a saddle-node bifurcation (Berglund & Gentz (2006, https://link.springer.com/content/pdf/10.1007/1-84628-186-5_3.pdf, Li et al. Physica D, 395, 7-14, (2019)), but we agree that this should be further clarified.

**Changes in manuscript:**

We will clarify this characteristic in the new appendix A and refer to it when presenting the results of this CESM experiment.

3. *The addition of tables would be helpful for readers to keep track of the many simulations (CESM, E-CCM, and reduced-order model) used in*

*this work. Please include details such as type (QE, branched from QE, steady state), duration, forcing characteristics, etc.*

**Author's reply:**

Agreed, a table of the CESM simulations is useful for the reader. For the E-CCM (with the many sensitivity experiments) such a table would be too extensive.

**Changes in manuscript:**

We will include an overview of the presented CESM simulations in the revised Methods section.

4. *For completeness, I would suggest including a brief discussion of the 5-box model (Section 2.2), such as a short description of the different boxes.*

**Author's reply:**

Suggestion followed.

**Changes in manuscript:**

We will expand the description of the E-CCM in the revision.

5. *Line 125: In theory, would it have been better to run the half-rate forcing experiment, branching off from a much lower value of freshwater forcing? Is a shorter overshoot (in $F_H$) the only difference you would expect to see, compared to existing results?*

**Author's reply:**

It would indeed be better to run the half-rate forcing experiment, starting further from the saddle-node bifurcation, but this is computationally too expensive. However, we expect similar results as the standard quasi-equilibrium remains closer to the statistical equilibria for lower $\overline{F_H}$. The differences become larger when moving closer to the saddle-node bifurcation.

**Changes in manuscript:**

We will further elaborate on the half-forcing rate simulation in the revision.

1. *Figures 2 and 5: The font size of the text in the subfigures is too small.*

   **Author's reply:**

   Panels c through n can share their $y$-axis to display them at a larger size.

   **Changes in manuscript:**

   We will revise Figures 2 and 5.

2. *Figure 5 (a,b): What causes the large overshoot in the backward quasi-equilibrium simulations after tipping back to the AMOC-on state?*

   **Author's reply:**

   The large overshoot is a model artefact and is not related to the dynamics of the E-CCM. This has been discussed in greater detail in van Westen et al. (2024b, https://doi.org/10.1175/JCLI-D-24-0060.1).

   **Changes in manuscript:**

   We will briefly mention this in the revision.

3. *Line 10: Typo: Should it be 'scenarios' (without the apostrophe)?*

   **Author's reply:**

   Agreed.

   **Changes in manuscript:**

   Will be corrected.

4. *Line 37: 'whether also this behavior' reads a bit awkward.*

   **Author's reply:**

   Agreed.

   **Changes in manuscript:**

   We will rewrite this as: '.... whether this behavior is also caused ...'

5. *Lines 166-167: Missing a 'to' in 'lower compared the'*

   **Author's reply:**

   Agreed.

   **Changes in manuscript:**

   Will be corrected.

6. *Line 387: Should it be 'built'?*

   **Author's reply:**

   Agreed.

   **Changes in manuscript:**

   Will be corrected.

7. *Line 391: What do you mean by 'with parameters somehow tuned'?*

   **Author's reply:**

   The E-CCM with sea-ice insulation effects was tuned to the CESM. These sea-ice effects were not considered in this manuscript, we only use the E-CCM in the temperature-varying and salinity-varying configuration.

   **Changes in manuscript:**

   We will remove this in the revision.

---

## Author Comment (AC2)

**MS-No.:** egusphere-2025-14

**Version:** Revision

**Title:** A Saddle-Node Bifurcation is Causing the AMOC Collapse in the Community Earth System Model

**Author(s):** René M. van Westen, Elian Vanderborght, Henk A. Dijkstra

**Point-by-point reply to reviewer**

**August 31, 2025**

We thank the reviewer for their careful reading and for the useful comments on the manuscript.

*This manuscript puts forward a number of arguments supporting AMOC collapse in the CESM model following the saddle-node bifurcation behaviour. The key arguments are that: there is a strict boundary of stability, a slower rate of quasi-equilibrium will lead to a faster tipping in forcing space, and that a simplified model shows the saddle-node bifurcation behaviour. The paper also demonstrates that warming under climate change can lead to tipping or lower levels of freshwater forcing for tipping. Finally, the paper demonstrates the sensitivity of the E-CCM model to locations of freshwater forcing.*

*The results are used to justify the continued importance of simplified and analytical model of the AMOC and provide guidance for the importance of model biases and the use of freshwater and climate forcing for AMOC tipping. I believe the paper may be suitable for publication in Earth System Dynamics providing the following comments are addressed.*

Major Comments

1. *The square root dependence needs more exploration. In the manuscript it is listed as the most important determinant of the saddle-node bifurcation but is then never discussed or justified to be ignored, and in Section 4 it is stated that Section 3.1 has shown it can't be demonstrated with these models, but Section 3 doesn't discuss the square root*

*dependence at all. The manuscript needs to discuss the square root dependence, how it would be analysed using these models, whether any information can be gained (i.e. are the current runs at least consistent with the square root behaviour within error, or are there reasons why the equilibrium runs wouldn't be expected to follow this behaviour?). How many runs would be required? Presumably 5 or 6 equilibrium runs near the threshold would be sufficient to analyse the shape?*

**Author's reply:**

Agreed, a better introduction for the square-root dependence is needed. This can already be mentioned in the introduction of the manuscript. The reduced model makes the square-root dependence explicit, which arises through the destabilising salt-advection feedback. It is difficult to asses how many equilibrium runs are needed to analyse the shape, as there is some uncertainty from atmospheric noise. Hence we rely on theoretical arguments that were presented in Section 4.

**Changes in manuscript:**

We will revise the text accordingly. In Section 4, we will also comment on the number of equilibrium runs needed.

2. *127-160: The overshoot under half-rate forcing seems statistically indistinguishable to that in the full-rate, so discussing the larger rate seems unrelated. The discussion of the feedbacks also seems unrelated as the comparison of feedbacks is not made between the rates. Finally, the key result drawn out in this section is that the collapse takes approximately 100 years each time, which is unrelated to the feedbacks. I think the relationship of the times and the forcing should be explored further while the feedbacks either need more explanation of their relevance or should be removed. Should the feedback analysis simply all be moved to section 4, where the importance of the feedbacks and their links to the simplified models is discussed in more detail.*

**Author's reply:**

We quantified the onset of the AMOC tipping event by using a break regression analysis. In the standard quasi-equilibrium simulation, this is found for $F_H$ between 0.522 to 0.533 Sv ($10^{th}$ and $90^{th}$ percentiles). Similarly, this is between 0.533 Sv to 0.536 Sv ($10^{th}$ and $90^{th}$ percentiles)

for the half-forcing simulation. This means that the AMOC tipping event in the half-forcing simulation is slightly later than the standard quasi-equilibrium simulation.

The text and related discussion around Figure 4 was confusing. The key point here was that the strength of the destabilising salt-advection feedback increases over centennial timescales ($\approx$ 200 years) prior to the onset of the AMOC tipping event. This is important to explain the differences in the overshoot between the standard and half-forcing CESM simulations, as well as the E-CCM results.

**Changes in manuscript:**

We will revise the text around Figure 4 accordingly. We will also quantify the onset of the AMOC tipping event for the two different quasi-equilibrium simulations using the break regression analysis.

3. *Greater discussion of the theory behind the analysis is needed. While other papers can be cited to justify arguments, more explanation is needed on why the saddle-node should follow a square-root behaviour, why the rates should lead to different rates of transition.*

**Author's reply:**

Agreed, we will expand the background theory of the analyses.

**Changes in manuscript:**

We will add a new appendix (Appendix A) to provide the required theory of the saddle-node bifurcation.

4. *Section 3.2 seems unrelated to the rest of the manuscript. The saddle-node behaviour in E-CCM has already been shown in a previous paper and can be referenced here. The response of the E-CCM model to the position of the freshwater forcing is interesting, but not discussed in the abstract, introduction, or summary of this paper and could be removed without impacting the rest of the paper. I suggest removing this section, potentially combining it with additional runs of the CESM model to compare the freshwater sensitivity and producing a separate manuscript discussing the sensitivity to location of freshwater forcing.*

*Alternatively, the relevance of this section to the rest of the manuscript should be justified.*

**Author's reply:**

The presented E-CCM results are relevant for the manuscript as they aid in the interpretation of the CESM results. We agree, however, that the E-CCM should be better introduced and tied to the CESM results. The E-CCM was mentioned in the abstract as the 'conceptual model' and also in the last paragraph of the introduction, but we will make this more explicit.

**Changes in manuscript:**

We will revise the text accordingly, where we will discuss the E-CCM results in greater detail and strengthen the link with the CESM results.

5. *Why was the region 20 and 50N chosen for the freshwater forcing when presumably the freshwater should be coming from further North? Is this a particularly sensitive area for freshwater forcing? This should be explained and justified as opposed to other regions. ? I believe this goes back to Stefan Rahmstorf's 1995 paper where the 20-50N region gives a clearer bifurcation than the Greenland forcing, but it would be helpful to clarify this here. (e.g. lines 57-59)*

**Author's reply:**

We used the hosing mask as in Hu et al. (2012, https://doi.org/10.1073/pnas.1116014109) and the the advantage of this 20°N to 50°N latitude band is that the deep convection areas are not directly impacted under the hosing. There are no large differences when varying the hosing region over the North Atlantic Ocean (Rahmstorf, 1996, https://link.springer.com/article/10.1007/s003820050144), as also shown through the hosing sensitivity analysis for the E-CCM.

**Changes in manuscript:**

We will extend the discussion on the hosing location in the revised Methods section.

1. *Title: I am not entirely convinced the strength of the title is justified and so could alter this to something weaker "A Saddle-Node Bifurcation may be causing the AMOC Collapse in the Community Earth System Model"?*

   **Author's reply:**

   Agreed.

   **Changes in manuscript:**

   Will be changed as suggested.

2. *10: no apostrophe needed in "scenario's"?*

   **Author's reply:**

   Agreed.

   **Changes in manuscript:**

   Will be corrected.

3. *13: Citation for the CMIP6 project/models?*

   **Author's reply:**

   Agreed.

   **Changes in manuscript:**

   We will include a reference to Eyring et al. (2016, https://doi.org/10.5194/gmd-9-1937-2016).

4. *19-20: Rephrase this sentence "The existence of the salt advection feedback is why the AMOC is labelled as a tipping point in the climate system"*

   **Author's reply:**

   Agreed.

**Changes in manuscript:**

We will incorporate this suggestion.

5. *37: "Whether this behaviour is also caused"*

   **Author's reply:**

   Agreed.

   **Changes in manuscript:**

   We will incorporate this suggestion.

6. *77: Why the "E-CCM"? Could you explain this to the reader otherwise it seems to appear out of nowhere, is it the extended-Cimatoribus climate model?*

   **Author's reply:**

   The acronym is defined as the Extended Cimatoribus Castellana Model.

   **Changes in manuscript:**

   We will define the acronym in the revision.

7. *Figure 2: The text and the Figures are very small, if they could be combined with shared axes or split perhaps with panels a and b at the top and the other 9 figures below them to allow more space.*

   **Author's reply:**

   Yes, we agree. The panels c through n can share their $y$-axis to display them at a larger size.

   **Changes in manuscript:**

   We will revise Figure 2 (and also Figure 5).

8. *I am confused by the relation between the steady states in panels a and b of Figure 2 and the right hand panels. Why is there no steady state behaviour for the 0.51 Sv case? Was this not left to equilibriate or were there other issues? Could you run a few steady state runs around 0.50*

*and 0.505 to specify the range of the bifurcation in more detail? Would this provide useful additional information*

**Author's reply:**

The red curves in panels 2h,n show that the AMOC collapse, indicative of the existence of a mono-stable AMOC regime for $\overline{F_H} = 0.51$ Sv. This means that the upper bound of the multi-stable regime is found for 0.495 Sv $\leq \overline{F_H} < 0.51$ Sv. Ideally, we would like to branch simulations from $\overline{F_H} = 0.495$ Sv with even smaller $\Delta\overline{F_H}$ increments to obtain a better estimate for the saddle-node bifurcation, but this is computationally too expensive. However, these sensitivity experiments can be conducted with the E-CCM.

**Changes in manuscript:**

We will expand the interpretation of the upper bound of the multi-stable regime in the Results section.

9. *112: Given this range of 0.495-0.510 does this significantly change later results, some of the changes are quite close to this? (see Major comment 2)*

**Author's reply:**

As was noted in the same line, the numerical results may slightly vary when using a different reference value, but do not change the interpretation.

**Changes in manuscript:**

No changes needed.

10. *119: Could you give more detail here about how the "tipping" is determined as the exact position seems very important and worth including in this manuscript even if discussed in more detail in another paper. (See Major comment 3)*

**Author's reply:**

We used a break regression analysis to find the onset of the AMOC tipping event at $F_H = 0.534$ Sv, with the $10^{\text{th}}$ and $90^{\text{th}}$ percentiles at $F_H = 0.533$ Sv and $F_H = 0.536$ Sv, respectively.

**Changes in manuscript:**

This will be included in the revision.

11. *Line 125: Uncertainty ranges on these numbers would be very helpful, there is clearly a lot of uncertainty in the reference value and probably also a large amount of uncertainty in where the AMOC collapses and so these values seem likely to be statistically indistinguishable. (See Major comment 2)*

   **Author's reply:**

   For the standard quasi-equilibrium simulation, the onset of the AMOC tipping event is found for $F_H$ between 0.522 to 0.533 Sv ($10^{\text{th}}$ and $90^{\text{th}}$ percentiles). For the half-forcing simulation, this is for $F_H$ between 0.533 Sv to 0.536 Sv ($10^{\text{th}}$ and $90^{\text{th}}$ percentiles). This means that the AMOC tipping event in the half-forcing simulation is slightly later than the standard quasi-equilibrium simulation.

   **Changes in manuscript:**

   We will add these uncertainties in the revision.

12. *Figure 4: If the comparison is trying to understand why the slower rate overshoots more than the faster overshoot, the faster overshoot should be included in this figure. This would allow clearer comparison*

   **Author's reply:**

   We are primarily interested in the strength of the different AMOC feedbacks, which are comparable between the different simulations. As was discussed in the manuscript (line 139 – 141), we expect that a critical $F_H$ was crossed causing the increase in the salt-advection feedback strength that eventually destabilises the AMOC. In the case of quasi-equilibrium or transient forcing, this critical $F_H$ depends on the forcing rate, as was made more explicit for the E-CCM (lines 182 – 184). We expect that the standard quasi-equilibrium simulation reached this critical $F_H$ earlier compared to the half-forcing quasi-equilibrium simulation.

**Changes in manuscript:**

We will rewrite these sentences to clarify the results presented in Figure 4.

13. *Lines 145-147: Does this suggest that 0.465 is a potential point at which the tipping might be initiated and that the other values are simply overshoots and the runs were not conducted enough times/ for long enough to find this tipping at the lower levels? Maybe the basic levels and overshoots discussed earlier in the manuscript should be adjusted to account for this.*

    **Author's reply:**

    This interpretation is indeed correct and we agree with the reviewer that this can be mentioned earlier in the manuscript. The AMOC eventually collapses in the branched simulation for $\overline{F_H} = 0.48$ Sv, meaning that a critical forcing value was already surpassed upon initialisation. Hence, the quasi-equilibrium also surpassed this critical forcing value and actually *undershoots* the saddle-node bifurcation. The apparent overshoot is related to inertia and the growth rate of AMOC feedbacks. This behaviour can be shown in greater detail for the E-CCM.

    **Changes in manuscript:**

    We will rewrite the text in Section 3 to clarify this.

14. *Lines 150-153: If all transitions take the same amount of time and the overshoot is just related to the rate of forcing vs the duration, surely this might be a general feature of other transitions? What transitions are we ruling out through this analysis? This is only helpful to determine a saddle-node bifurcation if it does not characterise other transitions.*

    **Author's reply:**

    This may indeed be a characteristic of several other bifurcations, so the behavior is not unique to the saddle-node bifurcation.

    **Changes in manuscript:**

    We will expand on this point in the revision.

15. *158-159: For the FovS, the variance increases up to the tipping but continues increasing after this, presumably if it actually tips in the saddle-node behaviour, the variance should decrease after the tipping and after settling in to the new behaviour? Is this true and should we expect this?*

    **Author's reply:**

    The $F_{\mathrm{ovS}}$ variance is larger in the AMOC off state compared to the AMOC on state (see Figure 2b). In the AMOC off state, there is only a shallow wind-driven overturning cell at 34°S that explains the larger $F_{\mathrm{ovS}}$ variance. This means that the $F_{\mathrm{ovS}}$ variance should increase after the AMOC tipping event.

    **Changes in manuscript:**

    No changes needed in the manuscript.

16. *Line 249-251: Why is it reasonable to ignore the sea-ice melt term when it is clearly one of the dominant terms in the model? Surely we want this reduced model to represent CESM not a box model and leaving out key terms such as this mean it is a poor representation when we know they are important.*

    **Author's reply:**

    The most important limitation is to express these contributions in terms of $S_{\rightleftarrows}$, which is not straightforward given the complex AMOC feedbacks (see discussion in lines $266 - 271$). Only a few terms in (2) can be directly related to $S_{\rightleftarrows}$.

    **Changes in manuscript:**

    We will extend the discussion in lines $249 - 251$ in the revision to clarify this.

17. *Equation 7: Could some of the freshwater not be stored in the Atlantic freshwater content rather than the variation terms? This assumes the Atlantic freshwater content must immediately return to equilibrium.*

**Author's reply:**

Correct, freshwater anomalies can also be stored in the Atlantic Ocean, but is not considered here as we express the freshwater balance in terms of $S_{\rightleftarrows}$.

**Changes in manuscript:**

We will mention the freshwater storage when discussing the limitations of the idealised model (near lines 266 – 271).

18. *Figure 8: Referring to the results as "Observations" seems unclear, the analysis is based on observed values but is not actual observations, the label in the legend could be changed for clarity.*

**Author's reply:**

Agreed, 'observed values' fit better here.

**Changes in manuscript:**

Figure 8 and text will be changed accordingly.

19. *316-326: This paragraph justifies multiple times that making these estimates is not useful or unrealistic. This paragraph should be removed or rephrased to emphasise that in the real world there are key differences which would lead to lower freshwater fluxes (e.g. the value of the FovS) but that there are other factors such as climate change and the location of forcing that make this unreliable.*

**Author's reply:**

The comparison with the real AMOC is still useful, in particular by demonstrating that the AMOC sensitivity ($\frac{\partial \text{AMOC}}{\partial F_H}$) is larger for the real AMOC than in the CESM. It also demonstrates that a positive $F_{\text{ovS}}$ bias results in an overly stable AMOC. We agree with the reviewer that the text was somewhat confusing and we need to rewrite this paragraph. We will also consider a different hosing location for the real AMOC, i.e., around the Greenland Ice Sheet.

**Changes in manuscript:**

We will rewrite this paragraph in the revision.

20. *386-388: You did not really show that this is not feasible for the CESM and need to justify this.*

    **Author's reply:**

    Yes, agreed. We will comment on this in the revision.

    **Changes in manuscript:**

    Text will be revised accordingly (see also Major comment 1).

21. *391: "parameters somehow tuned", probably just drop the word "somehow" here*

    **Author's reply:**

    The E-CCM with sea-ice insulation effects was tuned to the CESM. These sea-ice effects were not considered in this manuscript, we only use the E-CCM in the temperature-varying and salinity-varying configuration.

    **Changes in manuscript:**

    We will remove this in the revision.

22. *414-417: there is no section 4b, change this reference*

    **Author's reply:**

    This should be section 4.2, thank you for pointing this out.

    **Changes in manuscript:**

    Will be corrected.

---

## Author Response (AR1)

**MS-No.:** egusphere-2025-14

Version: Revision

Title: A Saddle-Node Bifurcation may be Causing the AMOC Collapse in the Community

Earth System Model

Author(s): René M. van Westen, Elian Vanderborght, Henk A. Dijkstra

We thank the Reviewers and the Editor for their careful reading and for the useful comments on the manuscript.

**Editor Report**

While both Reviewers underscore the merits of your submission, they also suggest some relatively substantial changes. I therefore return the manuscript to you for major revisions. As an overarching comment, I would encourage you to consider in full the reviewer comments. Where you propose a simple textual change in response to a comment asking for additional analysis, there should be a very convincing scientific argumentation for it.

**Author's reply:**

We have adequately addressed all the comments of the reviewers and have provided additional analyses (e.g., a new Appendix B) where needed.

**Point-by-point reply to Reviewer # 1**

General Comments: This paper investigates the mechanisms behind Atlantic Meridional Overturning Circulation (AMOC) collapse in the Community Earth System Model (CESM). The paper aims to demonstrate that the classical picture of a saddle-node bifurcation, as exhibited by box models, also holds for the AMOC collapse in the CESM. This is done by analyzing pre-industrial CESM simulations and comparing them with a conceptual model (E-CCM). The authors use physical arguments to demonstrate that the complex behavior of AMOC in the CESM can be approximated by a reduced-order model, in which a saddle-node bifurcation drives the AMOC collapse.

These results underscore the utility of idealized AMOC models and may help evaluate the effect of model biases on the AMOC stability landscape and for understanding AMOC responses under various climate change projections. The careful experimental design and related analyses, for simulations in CESM and E-CCM, are commendable. I think the paper may be suitable for publication after the following issues are satisfactorily addressed.

**Specific Comments**

1. The paper needs a discussion of the normal form of a saddle-node bifurcation and its properties (this could be in the appendix), given that the central aim of the work is to establish that the AMOC, as represented in CESM, has a saddle-node bifurcation. The strongest evidence for this is the square root dependence of the AMOC strength on the freshwater hosing in the reduced-order model developed from physical arguments. However, the paper does not explain why one should expect to see this square-root dependence.

**Author's reply:**

We agree that a better introduction of the saddle-node bifurcation and the associated square-root dependence is needed in the manuscript. We will include a new appendix (Appendix B) in the paper and refer to that in the introduction.

This is now mentioned in the introduction and throughout the revised manuscript. The normal form of the saddle-node bifurcation is  $\dot{x} = r - x^2$ , with more details in Appendix B.

2. Lines 391-394: Is the second argument in support of the existence of a saddle-node bifurcation in CESM qualitative? Does the 'two-times' faster transition in the 'half-rate' hosing experiment have any quantitative significance? Isn't this related to the inertia associated with the system? Lines 150-151 claim that this is a typical characteristic of transitions near saddle-node bifurcations. Could you explain this in more detail? I was unable to verify this claim by going through Kuehn 2011. In summary, I think the authors need to precisely clarify how this observation constitutes evidence for the existence of a saddle-node bifurcation.

**Author's reply:**

The two times faster transition in forcing space for the half rate experiment is a characteristic of a saddle-node bifurcation (Berglund & Gentz (2006, https://link.springer.com/content/pdf/10.1007/1-84628-186-5\_3.pdf, Li et al. Physica D, 395, 7-14, (2019)), but we agree that this should be further clarified.

**Changes in manuscript:**

This is now clarified in the revision by including Appendix B. Here we present the normal form of the saddle-node bifurcation and supports that transitions take place faster when lowering the freshwater flux forcing rate.

3. The addition of tables would be helpful for readers to keep track of the many simulations (CESM, E-CCM, and reduced-order model) used in this work. Please include details such as type (QE, branched from QE, steady state), duration, forcing characteristics, etc.

**Author's reply:**

Agreed, a table of the CESM simulations is useful for the reader. For the E-CCM (with the many sensitivity experiments) such a table would be too extensive.

**Changes in manuscript:**

The table is included in the revised Methods section.

4. For completeness, I would suggest including a brief discussion of the 5-box model (Section 2.2), such as a short description of the different boxes.

**Author's reply:**

Suggestion followed.

**Changes in manuscript:**

An extended description of the E-CCM is added in the revised manuscript.

5. Line 125: In theory, would it have been better to run the half-rate forcing experiment, branching off from a much lower value of freshwater forcing? Is a shorter overshoot (in FH) the only difference you would expect to see, compared to existing results?

**Author's reply:**

It would indeed be better to run the half-rate forcing experiment, starting further from the saddle-node bifurcation, but this is computationally too expensive. However, we expect similar results as the standard quasi-equilibrium remains closer to the statistical equilibria for lower  $\overline{F_H}$ . The differences become larger when moving closer to the saddle-node bifurcation.

**Changes in manuscript:**

We have elaborated on this point in the revision.

**Technical Comments**

1. Figures 2 and 5: The font size of the text in the subfigures is too small.

**Author's reply:**

Panels c through n can share their y-axis to display them at a larger size.

Figures 2 and 5 are revised accordingly.

2. Figure 5 (a,b): What causes the large overshoot in the backward quasiequilibrium simulations after tipping back to the AMOC-on state?

**Author's reply:**

The large overshoot (> 35 Sv) is a model artefact and is not related to the dynamics of the E-CCM. This has been discussed in greater detail in van Westen et al. (2024b, https://doi.org/10.1175/JCLI-D-24-0060.1).

**Changes in manuscript:**

This is now mentioned in the revision.

3. Line 10: Typo: Should it be 'scenarios' (without the apostrophe)?

**Author's reply:**

Agreed.

**Changes in manuscript:**

Corrected.

4. Line 37: 'whether also this behavior' reads a bit awkward.

**Author's reply:**

Agreed.

**Changes in manuscript:**

Was rewritten as: '.... whether this behavior is also caused ...'

5. Lines 166-167: Missing a 'to' in 'lower compared the'

**Author's reply:**

Agreed.

**Changes in manuscript:**

Corrected.

6. Line 387: Should it be 'built'?

**Author's reply:**

Agreed.

**Changes in manuscript:**

Corrected.

7. Line 391: What do you mean by 'with parameters somehow tuned'?

**Author's reply:**

The E-CCM with sea-ice insulation effects was tuned to the CESM. These sea-ice effects were not considered in this manuscript, we only use the E-CCM in the temperature-varying and salinity-varying configuration.

**Changes in manuscript:**

This was removed in the revision.

**Point-by-point reply to Reviewer # 2**

This manuscript puts forward a number of arguments supporting AMOC collapse in the CESM model following the saddle-node bifurcation behaviour. The key arguments are that: there is a strict boundary of stability, a slower rate of quasi-equilibrium will lead to a faster tipping in forcing space, and that a simplified model shows the saddle-node bifurcation behaviour. The paper also demonstrates that warming under climate change can lead to tipping or lower levels of freshwater forcing for tipping. Finally, the paper demonstrates the sensitivity of the E-CCM model to locations of freshwater forcing.

The results are used to justify the continued importance of simplified and analytical model of the AMOC and provide guidance for the importance of model biases and the use of freshwater and climate forcing for AMOC tipping. I believe the paper may be suitable for publication in Earth System Dynamics providing the following comments are addressed.

**Major Comments**

1. The square root dependence needs more exploration. In the manuscript it is listed as the most important determinant of the saddle-node bifurcation but is then never discussed or justified to be ignored, and in Section 4 it is stated that Section 3.1 has shown it can't be demonstrated with these models, but Section 3 doesn't discuss the square root dependence at all. The manuscript needs to discuss the square root dependence, how it would be analysed using these models, whether any information can be gained (i.e. are the current runs at least consistent with the square root behaviour within error, or are there reasons why the equilibrium runs wouldn't be expected to follow this behaviour?). How many runs would be required? Presumably 5 or 6 equilibrium runs near the threshold would be sufficient to analyse the shape?

**Author's reply:**

Agreed, a better introduction for the square-root dependence is needed. This can already be mentioned in the introduction of the manuscript. The reduced model makes the square-root dependence explicit, which arises through the destabilising salt-advection feedback. It is difficult

to asses how many equilibrium runs are needed to analyse the shape, as there is some uncertainty from atmospheric noise. Hence we rely on theoretical arguments that were presented in Section 4.

**Changes in manuscript:**

The text has been revised accordingly. In Section 4, we also commented on the number of equilibrium runs needed.

2. 127-160: The overshoot under half-rate forcing seems statistically indistinguishable to that in the full-rate, so discussing the larger rate seems unrelated. The discussion of the feedbacks also seems unrelated as the comparison of feedbacks is not made between the rates. Finally, the key result drawn out in this section is that the collapse takes approximately 100 years each time, which is unrelated to the feedbacks. I think the relationship of the times and the forcing should be explored further while the feedbacks either need more explanation of their relevance or should be removed. Should the feedback analysis simply all be moved to section 4, where the importance of the feedbacks and their links to the simplified models is discussed in more detail.

**Author's reply:**

We quantified the onset of the AMOC tipping event by using a break regression analysis. In the standard quasi-equilibrium simulation, this is found for  $F_H$  between 0.522 to 0.533 Sv (10th and 90th percentiles). Similarly, this is between 0.533 Sv to 0.536 Sv (10th and 90th percentiles) for the half-forcing simulation. This means that the AMOC tipping event in the half-forcing simulation is slightly later than the standard quasi-equilibrium simulation.

The text and related discussion around Figure 4 was confusing. The key point here was that the strength of the destabilising salt-advection feedback increases over centennial timescales ( $\approx 200$  years) prior to the onset of the AMOC tipping event. This is important to explain the differences in the overshoot between the standard and half-forcing CESM simulations, as well as the E-CCM results.

**Changes in manuscript:**

We completely rewritten the text around Figure 4. We also quantified the onset of the AMOC tipping event for the two different quasiequilibrium simulations using the break regression analysis.

3. Greater discussion of the theory behind the analysis is needed. While other papers can be cited to justify arguments, more explanation is needed on why the saddle-node should follow a square-root behaviour, why the rates should lead to different rates of transition.

**Author's reply:**

Agreed, we will expand the background theory of the analyses.

**Changes in manuscript:**

A new appendix (Appendix B) was added to provide the required theory of the saddle-node bifurcation.

4. Section 3.2 seems unrelated to the rest of the manuscript. The saddlenode behaviour in E-CCM has already been shown in a previous paper and can be referenced here. The response of the E-CCM model to
the position of the freshwater forcing is interesting, but not discussed
in the abstract, introduction, or summary of this paper and could be
removed without impacting the rest of the paper. I suggest removing
this section, potentially combining it with additional runs of the CESM
model to compare the freshwater sensitivity and producing a separate
manuscript discussing the sensitivity to location of freshwater forcing.
Alternatively, the relevance of this section to the rest of the manuscript
should be justified.

**Author's reply:**

The presented E-CCM results are relevant for the manuscript as they aid in the interpretation of the CESM results. We agree, however, that the E-CCM should be better introduced and tied to the CESM results. The E-CCM was mentioned in the abstract as the 'conceptual model' and also in the last paragraph of the introduction, but we will make this more explicit.

**Changes in manuscript:**

The text was revised accordingly and we provided a better justification for the E-CCM.

5. Why was the region 20 and 50N chosen for the freshwater forcing when presumably the freshwater should be coming from further North? Is this a particularly sensitive area for freshwater forcing? This should be explained and justified as opposed to other regions. ? I believe this goes back to Stefan Rahmstorf's 1995 paper where the 20-50N region gives a clearer bifurcation than the Greenland forcing, but it would be helpful to clarify this here. (e.g. lines 57-59)

**Author's reply:**

We used the hosing mask as in Hu et al. (2012, https://doi.org/10.1073/pnas.1116014109) and the the advantage of this 20°N to 50°N latitude band is that the deep convection areas are not directly impacted under the hosing. There are no large differences when varying the hosing region over the North Atlantic Ocean (Rahmstorf, 1996, https://link.springer.com/article/10.1007/s003820050144) as also shown

https://link.springer.com/article/10.1007/s003820050144), as also shown through the hosing sensitivity analysis for the E-CCM.

**Changes in manuscript:**

We extended the discussion on the hosing location in the revised Methods section.

**Minor Comments**

1. Title: I am not entirely convinced the strength of the title is justified and so could alter this to something weaker "A Saddle-Node Bifurcation may be causing the AMOC Collapse in the Community Earth System Model"?

**Author's reply:**

Agreed.

**Changes in manuscript:**

Suggestion followed.

2. 10: no apostrophe needed in "scenario's"?

**Author's reply:**

Agreed.

Corrected.

3. 13: Citation for the CMIP6 project/models?

**Author's reply:**

Agreed.

**Changes in manuscript:**

We included a reference to Eyring et al. (2016, https://doi.org/10.5194/gmd-9-1937-2016).

4. 19-20: Rephrase this sentence "The existence of the salt advection feedback is why the AMOC is labelled as a tipping point in the climate system"

**Author's reply:**

Agreed.

**Changes in manuscript:**

Suggestion followed.

5. 37: "Whether this behaviour is also caused"

**Author's reply:**

Agreed.

**Changes in manuscript:**

Suggestion followed.

6. 77: Why the "E-CCM"? Could you explain this to the reader otherwise it seems to appear out of nowhere, is it the extended-Cimatoribus climate model?

**Author's reply:**

The acronym is defined as the Extended Cimatoribus Castellana Model.

The acronym is now defined in the revision.

7. Figure 2: The text and the Figures are very small, if they could be combined with shared axes or split perhaps with panels a and b at the top and the other 9 figures below them to allow more space.

**Author's reply:**

Yes, we agree. The panels c through n can share their y-axis to display them at a larger size.

**Changes in manuscript:**

Figure 2 and Figure 5 are revised accordingly.

8. I am confused by the relation between the steady states in panels a and b of Figure 2 and the right hand panels. Why is there no steady state behaviour for the 0.51 Sv case? Was this not left to equilibriate or were there other issues? Could you run a few steady state runs around 0.50 and 0.505 to specify the range of the bifurcation in more detail? Would this provide useful additional information

**Author's reply:**

The red curves in panels 2h,n show that the AMOC collapse, indicative of the existence of a mono-stable AMOC regime for  $\overline{F_H} = 0.51$  Sv. This means that the upper bound of the multi-stable regime is found for 0.495 Sv  $\leq \overline{F_H} < 0.51$  Sv. Ideally, we would like to branch simulations from  $\overline{F_H} = 0.495$  Sv with even smaller  $\Delta \overline{F_H}$  increments to obtain a better estimate for the saddle-node bifurcation, but this is computationally too expensive. However, these sensitivity experiments can be conducted with the E-CCM.

**Changes in manuscript:**

We expanded the interpretation of the upper bound of the multi-stable regime in the Results section.

9. 112: Given this range of 0.495-0.510 does this significantly change later results, some of the changes are quite close to this? (see Major comment 2)

**Author's reply:**

As was noted in the same line, the numerical results may slightly vary when using a different reference value, but do not change the interpretation.

**Changes in manuscript:**

No changes needed.

10. 119: Could you give more detail here about how the "tipping" is determined as the exact position seems very important and worth including in this manuscript even if discussed in more detail in another paper. (See Major comment 3)

**Author's reply:**

We used a break regression analysis to find the onset of the AMOC tipping event at  $F_H = 0.534$  Sv, with the 10th and 90th percentiles at  $F_H = 0.533$  Sv and  $F_H = 0.536$  Sv, respectively.

**Changes in manuscript:**

This is now mentioned in the revision.

11. Line 125: Uncertainty ranges on these numbers would be very helpful, there is clearly a lot of uncertainty in the reference value and probably also a large amount of uncertainty in where the AMOC collapses and so these values seem likely to be statistically indistinguishable. (See Major comment 2)

**Author's reply:**

For the standard quasi-equilibrium simulation, the onset of the AMOC tipping event is found for  $F_H$  between 0.522 to 0.533 Sv (10th and 90th percentiles). For the half-forcing simulation, this is for  $F_H$  between 0.533 Sv to 0.536 Sv (10th and 90th percentiles). This means that the AMOC tipping event in the half-forcing simulation is slightly later than the standard quasi-equilibrium simulation.

These uncertainties were added in the revision.

12. Figure 4: If the comparison is trying to understand why the slower rate overshoots more than the faster overshoot, the faster overshoot should be included in this figure. This would allow clearer comparison

**Author's reply:**

We are primarily interested in the strength of the different AMOC feedbacks, which are comparable between the different simulations. As was discussed in the manuscript (line 139 - 141), we expect that a critical  $F_H$  was crossed causing the increase in the salt-advection feedback strength that eventually destabilises the AMOC. In the case of quasi-equilibrium or transient forcing, this critical  $F_H$  depends on the forcing rate, as was made more explicit for the E-CCM (lines 182 - 184). We expect that the standard quasi-equilibrium simulation reached this critical  $F_H$  earlier compared to the half-forcing quasi-equilibrium simulation.

**Changes in manuscript:**

The text in Section 3 was completely rewritten.

13. Lines 145-147: Does this suggest that 0.465 is a potential point at which the tipping might be initiated and that the other values are simply overshoots and the runs were not conducted enough times/ for long enough to find this tipping at the lower levels? Maybe the basic levels and overshoots discussed earlier in the manuscript should be adjusted to account for this.

**Author's reply:**

This interpretation is indeed correct and we agree with the reviewer that this can be mentioned earlier in the manuscript. The AMOC eventually collapses in the branched simulation for  $\overline{F_H} = 0.48$  Sv, meaning that a critical forcing value was already surpassed upon initialisation. Hence, the quasi-equilibrium also surpassed this critical forcing value and actually *undershoots* the saddle-node bifurcation. The apparent

overshoot is related to inertia and the growth rate of AMOC feedbacks. This behaviour can be shown in greater detail for the E-CCM.

**Changes in manuscript:**

The text in Section 3 was completely rewritten.

14. Lines 150-153: If all transitions take the same amount of time and the overshoot is just related to the rate of forcing vs the duration, surely this might be a general feature of other transitions? What transitions are we ruling out through this analysis? This is only helpful to determine a saddle-node bifurcation if it does not characterise other transitions.

**Author's reply:**

This may indeed be a characteristic of several other bifurcations, so the behavior is not unique to the saddle-node bifurcation.

**Changes in manuscript:**

We expanded the discussion on this point in the revision and is also demonstrated in Appendix B.

15. 158-159: For the FovS, the variance increases up to the tipping but continues increasing after this, presumably if it actually tips in the saddle-node behaviour, the variance should decrease after the tipping and after settling in to the new behaviour? Is this true and should we expect this?

**Author's reply:**

The  $F_{\text{ovS}}$  variance is larger in the AMOC off state compared to the AMOC on state (see Figure 2b). In the AMOC off state, there is only a shallow wind-driven overturning cell at 34°S that explains the larger  $F_{\text{ovS}}$  variance. This means that the  $F_{\text{ovS}}$  variance should increase after the AMOC tipping event.

**Changes in manuscript:**

No changes needed in the manuscript.

16. Line 249-251: Why is it reasonable to ignore the sea-ice melt term when it is clearly one of the dominant terms in the model? Surely we want

this reduced model to represent CESM not a box model and leaving out key terms such as this mean it is a poor representation when we know they are important.

**Author's reply:**

The most important limitation is to express these contributions in terms of  $S_{\rightleftharpoons}$ , which is not straightforward given the complex AMOC feedbacks (see discussion in lines 266 – 271). Only a few terms in (2) can be directly related to  $S_{\rightleftharpoons}$ .

**Changes in manuscript:**

This is now mentioned in the revision.

17. Equation 7: Could some of the freshwater not be stored in the Atlantic freshwater content rather than the variation terms? This assumes the Atlantic freshwater content must immediately return to equilibrium.

**Author's reply:**

Correct, freshwater anomalies can also be stored in the Atlantic Ocean, but this contribution on the freshwater content is assumed to be much smaller than the freshwater balance terms.

**Changes in manuscript:**

This limitation is now discussed for the reduced model.

18. Figure 8: Referring to the results as "Observations" seems unclear, the analysis is based on observed values but is not actual observations, the label in the legend could be changed for clarity.

**Author's reply:**

Agreed, 'observed values' fit better here.

**Changes in manuscript:**

Figure 8 and text was changed accordingly.

19. 316-326: This paragraph justifies multiple times that making these estimates is not useful or unrealistic. This paragraph should be removed or rephrased to emphasise that in the real world there are key differences

which would lead to lower freshwater fluxes (e.g. the value of the FovS) but that there are other factors such as climate change and the location of forcing that make this unreliable.

**Author's reply:**

The comparison with the real AMOC is still useful, in particular by demonstrating that the AMOC sensitivity  $(\frac{\partial \text{AMOC}}{\partial F_H})$  is larger for the real AMOC than in the CESM. It also demonstrates that a positive  $F_{\text{ovS}}$  bias results in an overly stable AMOC. We agree with the reviewer that the text was somewhat confusing and we need to rewrite this paragraph. We will also consider a different hosing location for the real AMOC, i.e., around the Greenland Ice Sheet.

**Changes in manuscript:**

This paragraph was rewritten in the revision.

20. 386-388: You did not really show that this is not feasible for the CESM and need to justify this.

**Author's reply:**

Yes, agreed. We will comment on this in the revision.

**Changes in manuscript:**

Text has been revised accordingly (see also Major comment 1).

21. 391: "parameters somehow tuned", probably just drop the word "somehow" here

**Author's reply:**

The E-CCM with sea-ice insulation effects was tuned to the CESM. These sea-ice effects were not considered in this manuscript, we only use the E-CCM in the temperature-varying and salinity-varying configuration.

**Changes in manuscript:**

This was removed in the revision.

22. 414-417: there is no section 4b, change this reference

**Author's reply:**

This should be section 4.2, thank you for pointing this out.

**Changes in manuscript:**

Corrected.

---

## Referee Report (RR1)

I would like to thank the authors for their detailed response to the previous round of comments, this is a much improved manuscript and I believe the clarity of the arguments is now much stronger. Overall, I believe that the main areas for improvement in the manuscript have now been address and only minor corrections now remain. I would support publication of this manuscript once these points have been addressed.

One significant conceptual point which remains for me is around Figure 9 and the cubic spline data. I believe that Fig 9a shows key information about the uncertainty of the minimum in the F\_ov. However, it is very unclear to me why fitting cubic splines is the right thing to do here, which it should match the functional form, and why it gives rise to the very uneven and bimodal structure in the finer resolution pdfs which I believe may be spurious structure based on a very limited dataset. I am concerned additionally by the spikes in the pdfs outside of the fitted range which are shown in the figure. These seem spurious and confusing to the reader, as I suspect they are not realistic and are artefacts of the fitting system. I think that this approach should be considered and potentially compared to approaches such as Gaussian Processes to understand if they are adding information beyond that in Fig 9a or whether they are adding structure which cannot be reasonably inferred from the available data and could be misleading. If this approach remains in the paper then I would like it to be explained in more detail, particularly around the spatial structure and bimodality. I do not believe this significantly impinges on the conceptual importance of the paper and I think Fig 9a is strong enough to stand on its own and makes the point clearly enough, but I think that the cubic splines section should be reconsidered.

**Minor Comments:**

- 1. Line 24: Perhaps ref Wood (2019) box model here as well
- 2. Line 63-65: With the section removed from the previous manuscript, this doesn't read as well and might now benefit from an "and" or some grammatical restructuring
- Line 165: It is somewhat unclear from this text whether the greater overshoot in forcing
  is expected at the slower rate or not, and if so then why this would be expected.
   Additional explanation here would be appreciated.
- 4. Figs 2 and 5 are improved but the text is still rather small, if this could be improved at all that would greatly improve them, perhaps the subplot titles could be placed onto the empty space in the subplots to allow more room for expansion? I think the figures are manageable as is, but improvements would be welcome
- 5. Line 209: Remind the reader that E\_A is the freshwater forcing, or potentially harmonise such that F\_H is freshwater forcing throughout the paper, although I understand that they are treated in different ways so this may be confusing.
- 6. Line 209-210: Ref Fig 5 to support this claim.
- 7. Line 284: "Similar as in the Stommel Model" should be "Similar to the Stommel Model"
- 8. Line 324: "precies" should be "precise"
- 9. Line 355-356: Some comment on how reasonable it is that these terms are small would be helpful, is there a citation or some analysis from models which suggests the order of magnitude of these terms and whether they can be safely ignored?
- 10. Fig 8: I think that "observed values" is still slightly misleading, perhaps it could be "CESM Model" and "Obs Model" just to be completely clear that this is idealised model data and not observations
- 11. Line 427: "Above-averaged" should be "above average"
- 12. Line 487: "Second argument" should be "The second argument"

- 13. Line 500: "loses resilience and making it more" should be "loses resilience, making it more"
- 14. Line 574 appears to be simply completing the square on the previous step, if the procedure from Faure Ragani is more significant than this or is imparting additional meaning to this procedure it would be helpful to state this explicitly

15.

---

## Author Response (AR2)

**MS-No.:** egusphere-2025-14

Version: Revision II

Title: A Saddle-Node Bifurcation may be Causing the AMOC Collapse in the Community

Earth System Model

Author(s): René M. van Westen, Elian Vanderborght, Henk A. Dijkstra

We thank the Reviewers again for their careful reading and for the useful comments on the revised manuscript.

**Point-by-point reply to Reviewer # 1**

I appreciate the addition of Appendix B on the normal form of the saddlenode bifurcation and the discussion of the nonautonomous case. The explanation for the half-rate forcing case is much improved now, and the corrected reference seems appropriate. I do not have any further comments and recommend acceptance for publication.

**Author's reply:**

Thank you!

**Point-by-point reply to Reviewer # 2**

I would like to thank the authors for their detailed response to the previous round of comments, this is a much improved manuscript and I believe the clarity of the arguments is now much stronger. Overall, I believe that the main areas for improvement in the manuscript have now been address and only minor corrections now remain. I would support publication of this manuscript once these points have been addressed.

One significant conceptual point which remains for me is around Figure 9 and the cubic spline data. I believe that Fig 9a shows key information about the uncertainty of the minimum in the  $F_{ov}$ . However, it is very unclear to me why fitting cubic splines is the right thing to do here, which it should match the functional form, and why it gives rise to the very uneven and bimodal structure in the finer resolution pdfs which I believe may be spurious structure based on a very limited dataset. I am concerned additionally by the spikes in the pdfs outside of the fitted range which are shown in the figure. These seem spurious and confusing to the reader, as I suspect they are not realistic and are artefacts of the fitting system. I think that this approach should be considered and potentially compared to approaches such as Gaussian Processes to understand if they are adding information beyond that in Fig 9a or whether they are adding structure which cannot be reasonably inferred from the available data and could be misleading. If this approach remains in the paper then I would like it to be explained in more detail, particularly around the spatial structure and bimodality. I do not believe this significantly impinges on the conceptual importance of the paper and I think Fig 9a is strong enough to stand on its own and makes the point clearly enough, but I think that the cubic splines section should be reconsidered.

**Author's reply:**

The cubic splines were motivated by the procedure outlined in van Westen et al. (2024, https://www.science.org/doi/full/10.1126/sciadv.adk1189). However, as pointed out by the reviewer, the limited number of statistical equilibria indeed gives rise to spurious behaviour (see thin lines in Figure 9a).

To obtain an unbiased estimate of the  $F_{\rm ovS}$  minimum, all  $F_{\rm ovS}$  combinations of the four statistical equilibria (i.e., 6,250,000 combinations) are considered, from which the frequency of the  $F_{\rm ovS}$  minimum per statistical

equilibrium is determined. The frequencies are: 1.1% ( $\overline{F_H} = 0.45 \text{ Sv}$ ), 21.7% ( $\overline{F_H} = 0.465 \text{ Sv}$ ), 43.2% ( $\overline{F_H} = 0.48 \text{ Sv}$ ) and 34.0% ( $\overline{F_H} = 0.495 \text{ Sv}$ ), with the weighted  $F_{\text{ovS}}$  minimum at  $F_H = 0.482 \text{ Sv}$ . This indeed confirms that the  $F_{\text{ovS}}$  minimum is most likely found for  $\overline{F_H} = 0.48 \text{ Sv}$ .

**Changes in manuscript:**

We kept the cubic spline procedure in the revised manuscript to demonstrate spurious behaviour for a low number of statistical equilibria. Revised Figure 9b now shows the four cumulative distribution functions of  $F_{\text{ovS}}$ , which are used to demonstrate that the  $F_{\text{ovS}}$  minimum is likely found for  $\overline{F_H} = 0.48$  Sv. The main text is changed accordingly.

**Minor Comments**

1. Line 24: Perhaps ref Wood (2019) box model here as well

**Author's reply:**

This is a relevant reference here.

**Changes in manuscript:**

Reference incorporated.

2. Line 63-65: With the section removed from the previous manuscript, this doesn't read as well and might now benefit from an "and" or some grammatical restructuring

**Author's reply:**

Agreed.

**Changes in manuscript:**

Sentences were rewritten.

3. Line 165: It is somewhat unclear from this text whether the greater overshoot in forcing is expected at the slower rate or not, and if so

then why this would be expected. Additional explanation here would be appreciated.

**Author's reply:**

This is hard to assess, as the sensitivity in overshooting/undershooting the saddle-node bifurcation depends on initialisation conditions, forcing rate, and hosing location. This was explicitly demonstrated for the E-CCM in Section 3.2 and Figure 6.

**Changes in manuscript:**

We comment on this point when introducing the half-forcing rate simulation.

4. Figs 2 and 5 are improved but the text is still rather small, if this could be improved at all that would greatly improve them, perhaps the subplot titles could be placed onto the empty space in the subplots to allow more room for expansion? I think the figures are manageable as is, but improvements would be welcome

**Author's reply:**

Suggestion followed.

**Changes in manuscript:**

The figure captions, the x- and y-ticks and the x- and y-labels are displayed at a larger size.

5. Line 209: Remind the reader that  $E_A$  is the freshwater forcing, or potentially harmonise such that  $F_H$  is freshwater forcing throughout the paper, although I understand that they are treated in different ways so this may be confusing.

**Author's reply:**

We prefer to use  $E_A$  for the E-CCM as the freshwater forcing is different compared to the CESM. A reminder to  $E_A$  is useful here.

**Changes in manuscript:**

Suggestion followed.

6. Line 209-210: Ref Fig 5 to support this claim.

**Author's reply:**

Agreed.

**Changes in manuscript:**

Suggestion followed.

7. Line 284: "Similar as in the Stommel Model" should be "Similar to the Stommel Model"

**Author's reply:**

Agreed.

**Changes in manuscript:**

Corrected.

8. Line 324: "precies" should be "precise"

**Author's reply:**

Agreed.

**Changes in manuscript:**

Corrected.

9. Line 355-356: Some comment on how reasonable it is that these terms are small would be helpful, is there a citation or some analysis from models which suggests the order of magnitude of these terms and whether they can be safely ignored?

**Author's reply:**

Figure 7 shows the  $F_{\rm gyre}$  and  $F_{\rm ovN}$  contributions up to  $F_H = 0.51$  Sv, which is beyond the critical value of  $\overline{F_H} = 0.48$  Sv (Figure 2f). This supports our claim that these terms remain reasonably small prior to the onset of the AMOC collapse. Thereafter, they have a considerable contribution (e.g., Figure 4) and is attributed to large-scale AMOC changes.

**Changes in manuscript:**

This is now discussed in the revised manuscript.

10. Fig 8: I think that "observed values" is still slightly misleading, perhaps it could be "CESM Model" and "Obs Model" just to be completely clear that this is idealised model data and not observations

**Author's reply:**

Agreed, observed model is more appropriate here.

**Changes in manuscript:**

We have revised Figure 8 and the text accordingly.

11. Line 427: "Above-averaged" should be "above average"

**Author's reply:**

Agreed.

**Changes in manuscript:**

Corrected.

12. Line 487: "Second argument" should be "The second argument"

**Author's reply:**

Agreed.

**Changes in manuscript:**

Corrected.

13. Line 500: "loses resilience and making it more" should be "loses resilience, making it more"

**Author's reply:**

Agreed.

**Changes in manuscript:**

Corrected.

14. Line 574 appears to be simply completing the square on the previous step, if the procedure from Faure Ragani is more significant than this or is imparting additional meaning to this procedure it would be helpful to state this explicitly

**Author's reply:**

Indeed, this is simply completing the square. This rewritten form motivates the scaling of variables.

**Changes in manuscript:**

We have rewritten these sentences.